# Direct observation of one-dimensional disordered diffusion channel in a chain-like thermoelectric with ultralow thermal conductivity

Jiawei Zhang [1✉], Nikolaj Roth[1], Kasper Tolborg [1], Seiya Takahashi[2], Lirong Song[1], Martin Bondesgaard[1], Eiji Nishibori [2] & Bo B. Iversen [1✉]

Structural disorder, highly effective in reducing thermal conductivity, is important in technological applications such as thermal barrier coatings and thermoelectrics. In particular, interstitial, disordered, diffusive atoms are common in complex crystal structures with ultralow thermal conductivity, but are rarely found in simple crystalline solids. Combining single-crystal synchrotron X-ray diffraction, the maximum entropy method, diffuse scattering, and theoretical calculations, here we report the direct observation of one-dimensional disordered $In^{1+}$ chains in a simple chain-like thermoelectric InTe, which contains a significant $In^{1+}$ vacancy along with interstitial indium sites. Intriguingly, the disordered $In^{1+}$ chains undergo a static-dynamic transition with increasing temperature to form a one-dimensional diffusion channel, which is attributed to a low $In^{1+}$-ion migration energy barrier along the $c$ direction, a general feature in many other TlSe-type compounds. Our work provides a basis towards understanding ultralow thermal conductivity with weak temperature dependence in TlSe-type chain-like materials.

[1] Center for Materials Crystallography, Department of Chemistry and iNANO, Aarhus University, DK-8000 Aarhus, Denmark. [2] Faculty of Pure and Applied Sciences and Tsukuba Research Center for Energy Materials Science (TREMS), University of Tsukuba, Tsukuba 305-8571, Japan. ✉email: jiaweizhang@chem.au.dk; bo@chem.au.dk

Structural disorder plays a crucial role in understanding the physical properties of a material[1]. Disorder and defects usually act as scattering centers for quasiparticles, such as electrons and phonons, influencing and governing the transport properties. In particular, the structural disorder is well known to scatter phonons and results in a lower thermal conductivity[2,3], an essential property that is fundamentally important in technological applications such as thermal barrier coatings and thermoelectric (TE) energy conversion. For the TE technology directly interconverting heat and electricity, reducing thermal conductivity is an essential strategy to improve the figure of merit $zT = \alpha^2\sigma T/\kappa$ of a TE material, where $\alpha$, $\sigma$, $T$, and $\kappa$ represent the Seebeck coefficient, electrical conductivity, absolute temperature, and thermal conductivity, respectively. Structural disorder with great potential in reducing thermal conductivity can be achieved in various ways. In addition to the substitutional disorder commonly used in alloys[3,4], the disorder can be realized intrinsically through vacancies, interstitial sites, off-center displacements, rattling atoms, or fast ionic diffusion within the unit cell in many state-of-the-art TE materials[5–9].

Understanding, characterizing, and modeling the structural disorder is of significant interest in TE materials with ultralow glass-like thermal conductivity, which poses great challenges to both theorists and experimentalists. Ultralow thermal conductivity is common in large complex crystal structures, whereas it is rare in simple crystal structures. However, in recent years many simple inorganic crystalline solids such as SnSe[10], $Tl_3VSe_4$[11], and $BaTiS_3$[12] have been discovered to show ultralow thermal conductivity approaching or even lower than the glass limit[13] for the amorphous and highly disordered solids. Theoretical models[11,14] recently developed have been quite successful in modeling ultralow, weak temperature-dependent thermal conductivities of crystalline solids through introducing a wave-like tunneling term for describing the disorder. However, very few detailed experimental characterizations on atomic disorder are available in these simple crystalline solids typically because of great challenges for experimentalists to probe subtle structural disorders. One class of such crystalline solids is the TlSe-type compounds. The TlSe structure type covers a rich variety of binary and ternary compositions, among which $TlInTe_2$, $TlGaTe_2$, TlSe, and InTe have been discovered as promising TE materials with ultralow thermal conductivity[15–18]. Ultralow thermal conductivity was typically attributed to strong anharmonic rattling induced by weakly bonded $In^{1+}$ or $Tl^{1+}$ atoms with lone pair electrons[16,17]. On the other hand, the atomic disorder, prone to appear for rattling atoms in oversize cages, should not be overlooked as it may also contribute to low thermal conductivity. In particular, the superionic conductivity[19,20] widely observed in this type of materials was suggested to be induced by the structural disorder and ion diffusion. However, no direct experimental evidence of the structural disorder or ion diffusion has been reported so far in any TlSe-type compound.

Here we combine multitemperature single-crystal synchrotron X-ray diffraction (SCSXRD) and the maximum entropy method (MEM) to obtain the electron density distribution for probing structural disorder in an archetypal TlSe-type compound InTe. A detailed structural analysis reveals a significant deficiency on the $In^{1+}$ site as the dominant intrinsic defect for understanding intrinsic p-type behavior, consistent with the theoretical defect calculations. Moreover, the structure contains two interstitial indium sites between the $In^{1+}$ sites, forming a one-dimensional (1D) disordered $In^{1+}$ chain along the $c$ axis. Interestingly, the MEM electron density at elevated temperatures and ab initio molecular dynamics simulations clearly show a static-dynamic transition of interstitial, disordered $In^{1+}$ ions, suggesting a 1D $In^{1+}$-ion diffusion/hopping pathway. We attribute such a

diffusion pathway to a very low $In^{1+}$-ion migration energy barrier along the $c$ direction, which is found to be a general feature in many other TlSe-type compounds. The local correlations in the 1D $In^{1+}$ chains are examined through single-crystal diffuse X-ray scattering, and it is found that the static-dynamic transition is related to the degree of local order in the chains. The direct visualization of a 1D disordered diffusion channel not only accounts for the reported superionic conductivity but also provides a basis for understanding ultralow thermal conductivity and its peculiar, weak temperature dependence in InTe and other TlSe-type compounds.

## Results

**Ideal crystal structure and thermal conductivity**. Despite simple crystal structure with relatively light mass, InTe has been reported to show ultralow thermal conductivity values of ~0.7–0.4 W m$^{-1}$ K$^{-1}$ at 300–700 K[16], much lower than those in many well-known binary tellurides such as PbTe and $Bi_2Te_3$[3]. InTe, crystallizing in the TlSe-type structure with the space group $I4/mcm$[21], is generally described by the formula $In^{1+}In^{3+}Te_2^{2-}$. The $In^{3+}$ ions are tetrahedrally coordinated to $Te^{2-}$ ions forming $(InTe_2)^-$ chains along the $c$ axis, while the $In^{1+}$ ions with $5s^2$ lone pair electrons are weakly bound to a cage-like system of eight Te atoms with the square antiprismatic arrangement (Fig. 1a, b). Thermal displacement parameters of $In^{1+}$ ions were found by Hogg and Sutherland[21] to be very large and anisotropic with the maximum vibration along the $c$ axis. In this study, we synthesized large InTe single crystals using the vertical Bridgman method. Energy-dispersive spectroscopy (EDS) mapping was conducted using scanning transmission electron microscopy (STEM) and the result confirmed a uniform distribution of In and Te in the crystal (Supplementary Fig. 1). The chemical composition of the single crystal was determined to be $x = 0.98 \pm 0.01$ in $In_xTe$ by inductively coupled plasma optical emission spectrometry (ICP-OES) (Supplementary Table 1). Due to the weak interactions between $(InTe_2)^-$ chains, the crystal tends to cleave along the $(hh0)$ planes (Fig. 1c). The intersection line of two perpendicular cleavage planes $(hh0)$ of A and B was used to determine the crystallographic $c$ direction (Supplementary Fig. 2). With the identified crystal orientation, we then measured the thermal conductivity from 2 to 723 K for the InTe crystals along the $c$ direction (see Fig. 1d). The measured low-temperature thermal conductivity data along the $c$ direction show a very good agreement with those reported in ref. [22]. By comparing with the reported data along the [110] direction in ref. [23], it is found that the thermal conductivity of InTe single crystal is clearly anisotropic at room temperature but becomes less anisotropic or even nearly isotropic at high temperatures. A peculiar feature of the thermal conductivity is found with a nearly temperature-independent behavior ($\sim T^{-0.1}$) at low temperatures of ~25–80 K, where the extrinsic scattering should be more important. Even with increasing temperature well above the Debye temperature (~120 K, Supplementary Table 2), where the intrinsic phonon-phonon scattering is expected to be dominant, the temperature dependence of thermal conductivity ($\sim T^{-0.69}$) still deviates from $T^{-1}$. Such abnormal behavior, also observed in ref. [22], is typically an indication of the extrinsic scattering induced by structural disorder. In this work, we focus on a proper description of the crystal structure as a basis for a profound understanding of the atomic disorder in thermoelectric InTe.

**Disordered $In^{1+}$ atoms and intrinsic defects**. High-resolution SCSXRD data were used for the detailed crystal structure determination of InTe. We conducted the SCSXRD measurements at the BL02B1 beamline[24] from SPring-8 using a high photon

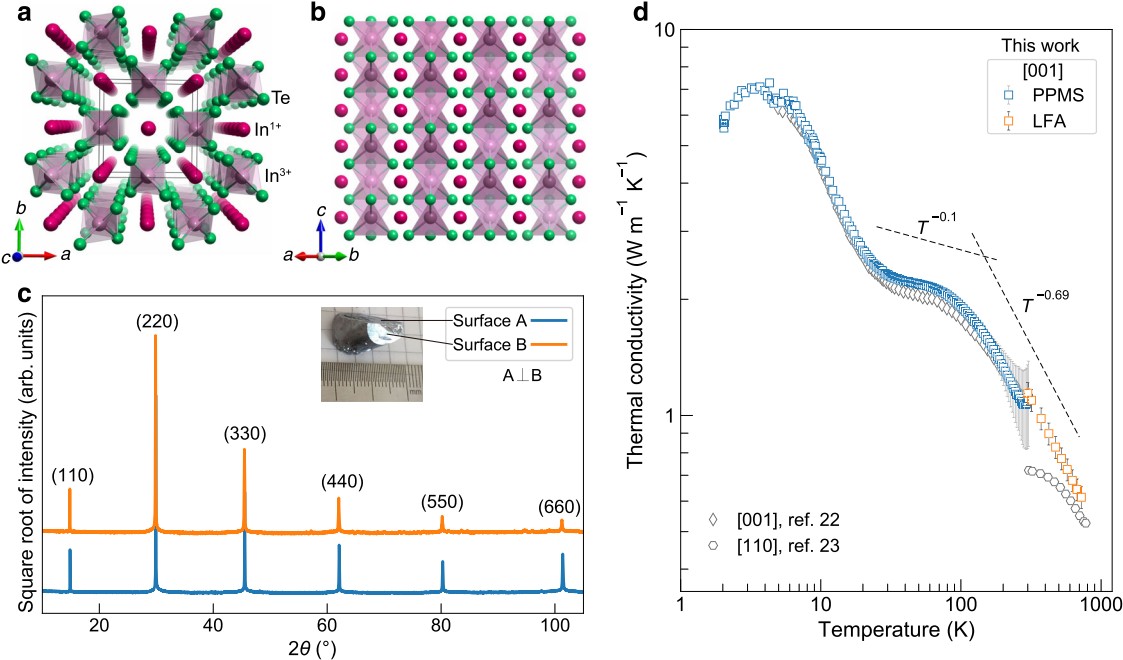

**Fig. 1 Ideal crystal structure and thermal conductivity of InTe single crystal. a, b** Illustration of the reported ideal crystal structure of InTe along the *c* axis (**a**) and the [110] direction (**b**). **c** XRD patterns of the cleavage surfaces of InTe single crystal for the determination of the crystallographic directions. The intersection line of the two perpendicular cleavage planes (*hh*0) determines the *c* axis. The inset photo shows the cleaved crystal with two perpendicular cleavage surfaces of A and B. **d** Measured temperature-dependent thermal conductivity of InTe single crystal along the [001] direction. The reported thermal conductivity data along [001] and [110] directions of InTe single crystal from refs. [22,23] are plotted for comparison. The PPMS and LFA denote the Physical Property Measurement System and Laser Flash Apparatus for the low- and high-temperature thermal conductivity data measurements, respectively.

energy of 50.00 keV. The data at 25 K were measured with a high resolution of $\sin\theta/\lambda < 1.67$ Å$^{-1}$ on a small high-quality single-crystal extracted from the as-grown large crystal. A total of 89,683 reflections were collected and reduced to 2723 unique reflections with a high redundancy of 32.9 and $R_{merge} = 4.13\%$ (Supplementary Table 3). The structure at 25 K was solved with lattice parameters of $a = 8.3685(3)$ and $c = 7.1212(2)$ Å in $I4/mcm$ space group. Accurate structure factors obtained from the SCSXRD data were used for MEM calculations. The MEM can provide a non-biased reconstruction of the most probable electron density from scaled and phased structure factors[25,26], making it a powerful tool to study the location of atoms[27], structural disorder[5,28], and migration pathways of ions[29,30].

Combining the MEM electron density and structure refinements, a precise structure model can be derived for understanding the atomic disorder (Fig. 2a, b). A three-dimensional (3D) MEM electron density map of InTe at 25 K (Fig. 2a) clearly reveals two small, nonequivalent additional regions of electron density between the In$^{1+}$ atoms along the *c* direction. The two-dimensional (2D) slice and 1D profile of the MEM density further confirm two small electron density maxima between the In$^{1+}$ atoms (Fig. 2c, d, and Supplementary Figs. 3–5), suggesting two interstitial sites with low occupancy in addition to the three main atomic sites. The localized nature of the extra electron density with distinct peaks along the *c* direction demonstrates a clear 1D static disordered feature. By analyzing the peak positions with crystal symmetry, two interstitial sites were determined to be 8 f (0.5 0.5 0.5463(4)) and 8 f (0.5 0.5 0.6369(4)). If interstitial indium atoms of In$_i$(1) and In$_i$(2) are allocated to the two additional sites, a two interstitial model can be obtained (see Fig. 2b and Supplementary Table 4).

To better elaborate the structure determination, we conducted and compared the refinements of different structure models with intermediate steps towards the two interstitial model (see Table 1

and Supplementary Tables 5–7). We started with the reported simple full occupancy model[21] with three main atomic sites fully occupied, then moved to the vacancy model with the indium vacancy allowed, and finally explored the two interstitial model with two interstitial indium atoms added. Notably, we found that each intermediate step greatly improves the agreement with the experimental SCSXRD data as the $R_F$ value drops remarkably from 2.54% to 2.06% to 1.33%, the $wR_F$ value from 5.54% to 4.14% to 2.15%, and the goodness-of-fit (GoF) from 3.02 to 2.26 to 1.37. In addition to the significantly improved fit to the experimental diffraction data, only the two interstitial model shows a composition of In$_{0.98}$Te consistent with the experimental value measured by ICP-OES.

The new structure determination paves the way to understand the dominant intrinsic defects and persistent p-type behavior of InTe. In contrast to the nearly fully occupied In$^{3+}$ site, our structure refinements clearly show a significant vacancy on the In$^{1+}$ site with only ~90% occupancy (Table 1). Moreover, the distances between In$^{1+}$ and In-interstitial sites are generally too small (<2.2 Å) so that it is unlikely for indium atoms to occupy these sites at the same time, suggesting the formation of Frenkel pairs. The dominant In$^{1+}$ vacancy along with small proportions of the Frenkel pairs thereby reasonably describes the observed structural disorder. To further elucidate the experimental observation, we conducted formation energy calculations of native defects in InTe using density functional theory. As depicted in Fig. 2e, the vacancy on the In$^{1+}$ site is the dominant native defect showing a much lower formation energy than other intrinsic defects including the In$^{3+}$ vacancy, Te vacancy, and antisite defects. The easier formation of the In$^{1+}$ vacancy may be attributed to very weak adjacent bonds of In$^{1+}$ atoms, showing large nearest-neighbor distances of 3.56 and 3.54 Å, respectively with In$^{1+}$ and Te atoms. The dominant native defect of the In$^{1+}$ vacancy, normally negatively charged, is known to pin the Fermi

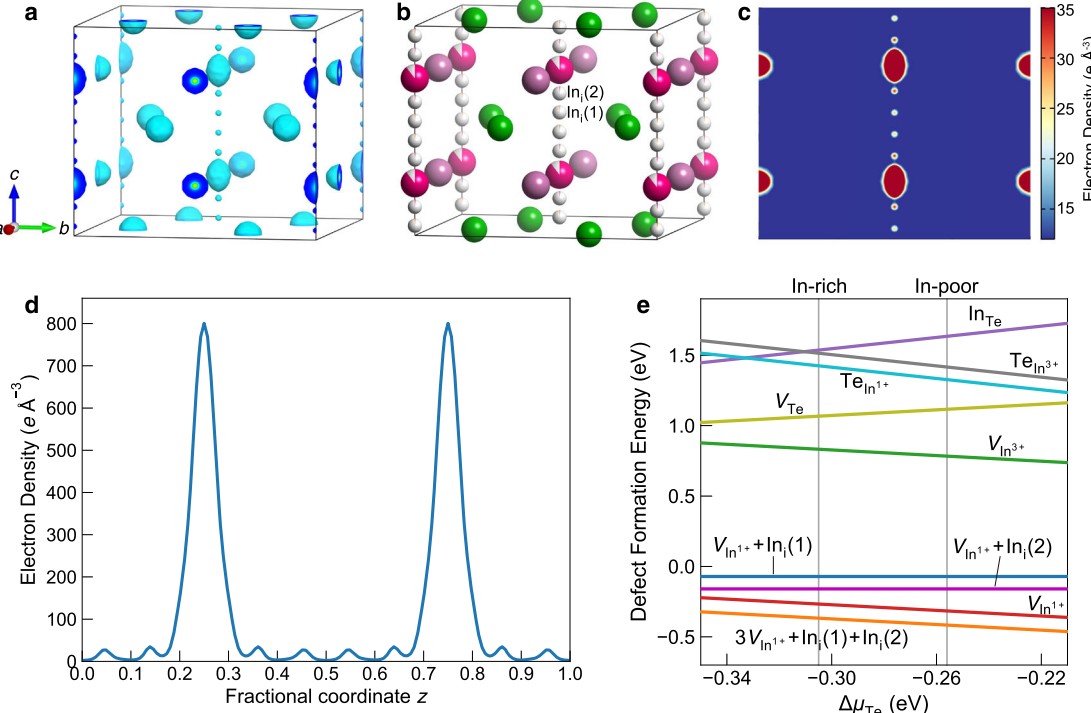

**Fig. 2 Disordered indium atoms and intrinsic defects in InTe revealed by the MEM electron density at 25 K. a** 3D electron density surface of InTe calculated by the MEM using high-resolution SCSXRD data at 25 K. The isosurface value is 15 $e$ Å$^{-3}$. **b** Two interstitial structure model based on the MEM electron density for InTe. **c** 2D MEM electron density map on the (100) plane at $x = 0.5$ for InTe. **d** 1D MEM electron density profile through the In$^{1+}$ atoms along the [001] direction. The origin of the line plot is placed at ($x = 0.5$, $y = 0.5$, $z = 0$). **e** Theoretical formation energies of native defects in InTe.

**Table 1 Comparison of different structure models for InTe with intermediate steps to the two interstitial model at 25 K.**

| Site | Site occupancy (fractional) | | | | $R_F$ | $wR_F$ | GoF | $x$ in In$_x$Te |
|---|---|---|---|---|---|---|---|---|
| | In$^{1+}$ | In$^{3+}$ | Te | In$_i$ | (%) | (%) | | |
| Multiplicity | (4a) | (4b) | (8 h) | (8 f) | | | | |
| Full occupancy model | 1 | 1 | 1 | 0 | 2.54 | 5.54 | 3.02 | 1 |
| Vacancy model 1 | 0.901 (2) | 0.999 (2) | 1 | 0 | 2.06 | 4.14 | 2.26 | 0.95 |
| Vacancy model 2 | 0.901 (2) | 1 | 1 | 0 | 2.06 | 4.14 | 2.26 | 0.95 |
| Two interstitial model | 0.895 (1) | 1 | 1 | 0.031 (1) | 1.33 | 2.51 | 1.37 | 0.98 |
| ICP-OES | | | | | | | | 0.98 (1) |

Note: The reliability factors $R_F$ and $wR_F$ based on structure factors as well as goodness-of-fit (GoF) describe the agreement between the structure model and experimental diffraction data. Only the two interstitial model gives the composition agreeing well with the experimental value measured by the inductively coupled plasma optical emission spectrometry (ICP-OES). The occupancy of In$_i$ represents the sum of site occupancies of the two indium interstitials.

level close to the valence band maximum, which therefore explains the persistent p-type behavior of InTe reported in previous studies[16,23].

In comparison with the In$^{1+}$ vacancy, the formation of Frenkel pairs including the experimental indium-interstitials alone is energetically unfavorable owing to a bit higher formation energies (Fig. 2e), which is expected due to the resulting small In-In distances. However, on the basis of the In$^{1+}$ vacancy, the formation of Frenkel pairs becomes more favorable with lower formation energy, indicating that the formation of Frenkel pairs in InTe is vacancy-mediated, consistent with the experimental observation. This may typically be ascribed to the vacancy-induced void space that helps avoid small In-In distances during the formation of Frenkel pairs.

**Temperature-driven 1D diffusive In$^{1+}$ channel**. To explore the temperature dependence of the disordered indium in InTe, we

further conducted MEM calculations based on the SCSXRD data at 100–700 K (Supplementary Table 8 and Supplementary Figs. 6–10). Similarly, the SCSXRD data at 100–700 K were collected with high resolutions of $(\sin\theta/\lambda)_{max} = 0.97$–$1.83$, as well as high redundancy of 28.8-88.0 (Supplementary Table 3). The 3D and 2D MEM electron density maps at various temperatures of 25–700 K are plotted in Fig. 3a, b for comparison. Strikingly, with increasing temperature, electron densities on the In$^{1+}$ and interstitial In sites become delocalized and eventually form a smooth, continuous 1D diffusion channel through In$^{1+}$ atoms along the $c$ direction. In Fig. 3c we compare the normalized 1D MEM electron density profile through In$^{1+}$ atoms along the [001] direction at 25–700 K (see also Supplementary Figs. 11 and 12). At 25 and 100 K, the electron densities with distinct peaks on the two interstitial In sites are generally well separated, though the electron densities of the In$^{1+}$ and the nearest interstitial In$_i$(2) are partially connected. With increasing temperature from 100 to 200 K, there are two notable changes in the 1D electron density

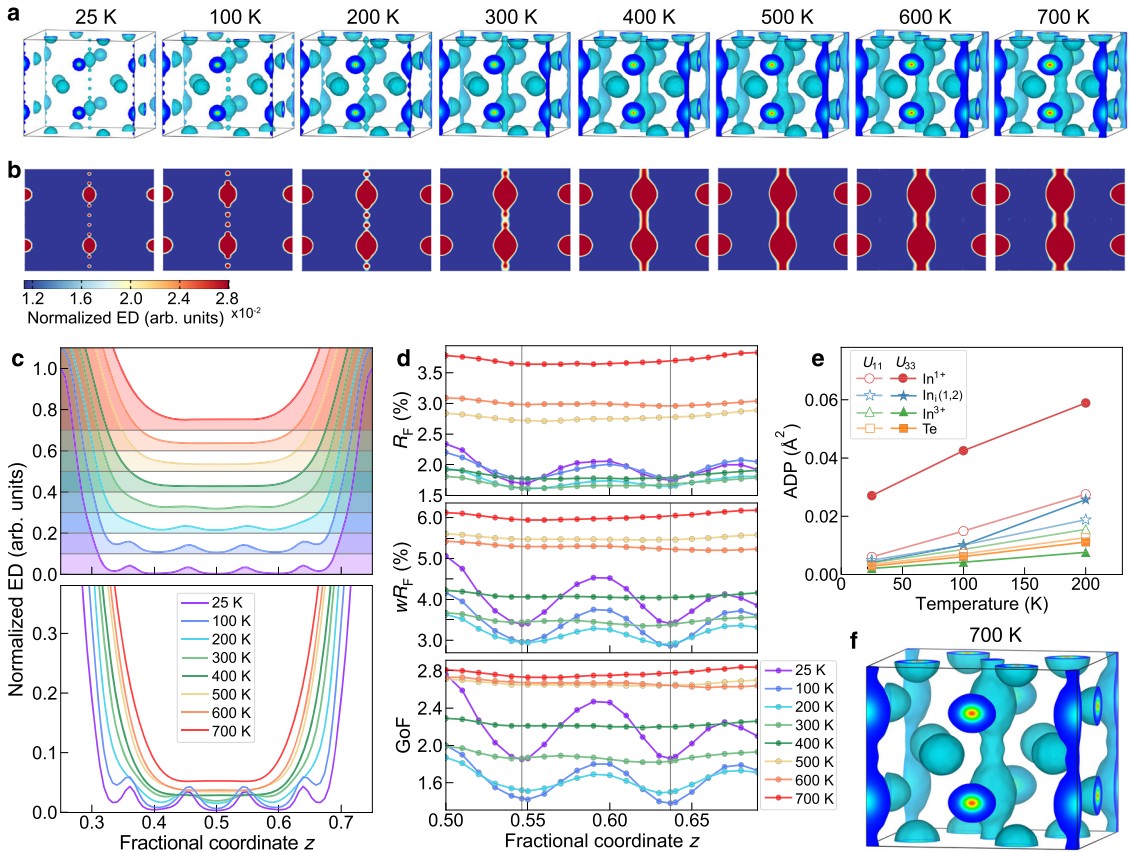

**Fig. 3 Temperature-driven 1D diffusive, disordered In$^{1+}$ channel in InTe. a** 3D MEM electron density maps at 25-700 K. The isosurface values are set at 1.45% and 2.35% of the peak electron density values of the In$^{1+}$ sites for the 25-600 K and 700 K MEM maps, respectively. **b** 2D MEM electron density maps at 25-700 K on the (100) plane at $x = 0.5$. The 2D MEM density maps at 25-600 K and 700 K are normalized, respectively, with one times and 1.58 times of the maximum values at the In$^{1+}$ sites. The ED denotes the electron density. **c** 1D MEM electron density profile through the In$^{1+}$ ions along the [001] direction at 25–700 K. The EDs at different temperatures are normalized for better comparison. **d** The $R_F$, $wR_F$, and GoF values for the constrained structure refinements with one interstitial indium moving along the $c$ axis between In$^{1+}$ sites. Two light gray vertical lines represent the locations of the two interstitial indium sites. **e** Anisotropic atomic displacement parameters (ADPs) of InTe refined with the two interstitial model. **f** Enlarged 3D MEM electron density map of InTe at 700 K with an isosurface value of 1.01 $e$ Å$^{-3}$.

profile. One is that the In$^{1+}$ density expands and merges with the electron density of the In$_i$(2), resulting in the disappearance of the peak on the interstitial site In$_i$(2). The other is the onset of the electron density connection between the In$^{1+}$, In$_i$(1), and In$_i$(2) through the $c$ direction (see also Supplementary Fig. 13). The two clear changes in electron density are likely the origin of the kink observed on the resistivity *versus* temperature curve between 100 and 200 K (Supplementary Fig. 14). The kink in resistivity data has been consistently observed in both single-crystalline[22] and polycrystalline[31] InTe at low temperatures and is thus not due to a measurement artifact. Finally, as the temperature increases further (≥300 K) the electron density maxima of the In$_i$(1) are smeared out and flattened, forming a continuous 1D channel connecting In$^{1+}$ atoms along the $c$ direction.

In Fig. 3d we show the temperature-dependent constrained structure refinements with one interstitial indium manually moving along the 1D channel between the In$^{1+}$ main sites. At temperatures of 25–200 K, the $R_F$, $wR_F$, and GoF values all show two distinct local minima, the location of which exactly correspond to the two indium interstitial sites. This signifies the generally static feature of the two interstitial indium sites. In contrast, at elevated temperatures of 300–700 K the $R_F$, $wR_F$, and GoF values generally become nearly constant with moving the interstitial indium between the In$^{1+}$ sites along the $c$ direction, suggesting nearly equal probability for the occupation of the

interstitial indium. This result along with the MEM densities indicates a static-dynamic transition of the interstitial indium positions. The dynamic behavior of the interstitial indiums makes the structure refinements with the two interstitial model only feasible up to 200 K (Supplementary Table 7). A sudden increase in the thermal vibration along the $c$ axis ($U_{33}$) of the interstitial indiums between 100 and 200 K shown in Fig. 3e suggests the onset of the dynamic motion along the $c$ axis, consistent with the MEM density at 200 K (Fig. 3c and Supplementary Fig. 13). It should be noted that we are not able to determine whether this dynamic behavior is induced by the thermal disorder or the dynamically positional disorder since the time-space averaged MEM density is not deconvoluted with the thermal smearing effect. Nevertheless, the continuous 1D MEM density channel above 200 K clearly reveals the probability for In$^{1+}$ to diffuse or hop between the In$^{1+}$ sites, indicating the 1D In$^{1+}$-ion hopping/ diffusion pathway along the $c$ direction (Fig. 3a, f). The In$^{1+}$-ion diffusion/hopping probability increases with increasing temperatures, but is generally not very high as the MEM electron density value of the diffusion channel even at 700 K is about 5.3% of the maximum peak value of the main In$^{1+}$ site (Fig. 3c, f). The ion diffusion/hopping in InTe is clearly weaker than those in conventional superionic conductors[5,9,29], which might pose a challenge to probe the signature of superionic conductivity in InTe. An external driving force such as a higher temperature, a

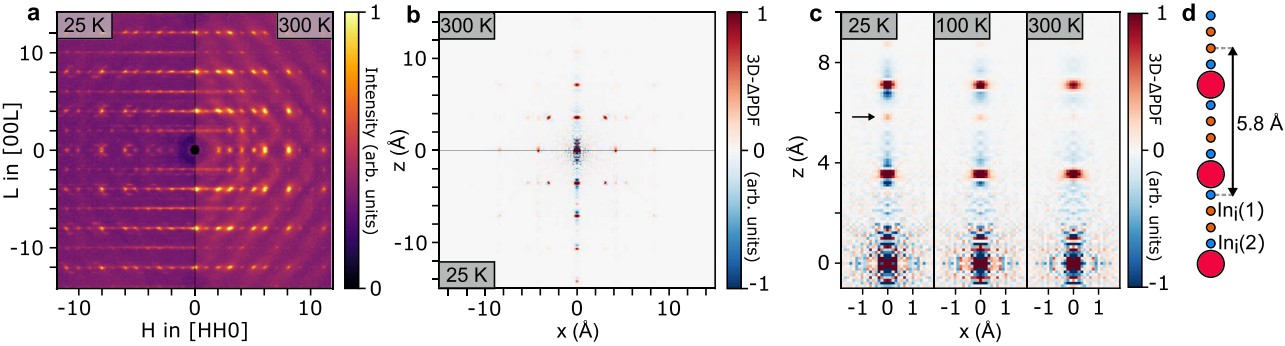

**Fig. 4 Local ordering in 1D In$^{1+}$ chains. a** Measured diffuse X-ray scattering in the *HHL* plane at 25 and 300 K. **b** 3D-ΔPDF in the *x0z* plane at 25 and 300 K. **c** Zoom on features in the 3D-ΔPDF along *z* at 25, 100 and 300 K. **d** Illustration of the 1D chain with the characteristic distance between interstitial sites marked, corresponding to the positive peak marked in (**c**) by a black arrow.

large electric field[20], or the increased In$^{1+}$ vacancy may be required to enhance the diffusion/hopping probability. The In$^{1+}$ diffusion/hopping may be associated with very large thermal motions of the In$^{1+}$ and interstitial indiums along the *c* direction (Fig. 3e and Supplementary Fig. 15). In particular, exceptionally large $U_{33}$ values of the In$^{1+}$ atom at elevated temperatures do not extrapolate to zero at 0 K, another clear indication of atomic disorder along the *c* direction[32].

The local structural ordering of vacancies and interstitial sites in the 1D chains was investigated through diffuse X-ray scattering, shown in Fig. 4a (see also Supplementary Fig. 16). At 25 K the diffuse scattering forms 2D planes for even values of *L*, seen as lines in the figure, indicative of strong correlations along the *c* direction, and weak correlations in other directions[33]. At 300 K the planes have disappeared and been replaced by more 3D features with strong maxima at the Bragg peak positions, a typical indication of thermal diffuse scattering from correlated vibrations. This is seen in more detail using the three-dimensional difference pair distribution function (3D-ΔPDF), shown in Fig. 4b (see also Supplementary Fig. 17). The 3D-ΔPDF shows the local correlations in disordered crystals[34–37]. It is a map showing which interatomic vectors are more or less present in the real structure compared to the average crystal structure. Positive features show vectors, which separate atoms more frequently in reality and negative features show vectors separating fewer atoms. The 3D-ΔPDF at 25 K shows strong correlations along *z*, which become weaker at 300 K, showing the vacancies and interstitials to be strongly correlated along the 1D chains at low temperatures, but less so at higher temperatures. This is shown in more detail along the *z* direction in Fig. 4c. At *x*, *y*, *z* = 0 there is noise due to Fourier ripples. For vectors equal to ½*c* and 1*c* (*z* equal to 3.56 and 7.1 Å), there are positive peaks, indicating that In atoms will most often be separated by these vectors, as they would in the ideal structure. Around these positive peaks, there are negative signals, indicating vectors that do not separate In atoms in the real structure. At low temperature the negative signal around positive peaks is asymmetric, a typical indication of local relaxations from atoms moving slightly towards neighboring vacancies[34,37]. At 25 K a positive peak is found at 5.8 Å, corresponding to a vector separating an In$_i$(1) and an In$_i$(2) interstitial, as illustrated in Fig. 4d, suggesting that the interstitial Frenkel pairs tend to be separated by this distance. At 300 K the asymmetry in the negative signal and the Frenkel pair peak have disappeared, indicating that the positions of vacancies and interstitials are less correlated at high temperatures consistent with the 1D In$^{1+}$-ion diffusion/hopping behavior.

The experimental observation of the 1D In$^{1+}$-ion diffusion pathway is further confirmed by the ab initio molecular dynamics simulation. As illustrated in Fig. 5a, b, and Supplementary Fig. 18,

the trajectories of the In$^{1+}$ ions for the simulation at 700 K show a clear hopping/diffusion behavior along the *c* direction. We note that only a limited number of In$^{1+}$ ions show 1D diffusion/hopping behavior in the simulated supercell, indicating that the probability of the In$^{1+}$-ion diffusion is not very significant consistent with the experimental MEM electron density. To understand the 1D diffusion/hopping pathway of the In$^{1+}$ in InTe, we further conducted nudged elastic band calculations to estimate the energy barriers for the vacancy-mediated In$^{1+}$-ion migration. Three possible indium ion migration pathways along the [001], [110], and [100] directions were considered (Fig. 5c). Notably, the In$^{1+}$ migration barrier along the [001] direction is significantly lower than those along the other two directions (Fig. 5d), which reasonably explains the 1D diffusion channel along the *c* direction in the MEM electron density. The flat energy landscape with a very low energy barrier may be attributed to the weak atomic interaction along the *c* direction.

## Discussion

Interstitial, disordered, diffusive atoms with large thermal motions are known to be a highly effective mechanism for reducing thermal conductivity. One remarkable example is $Zn_4Sb_3$, which shows extremely low thermal conductivity due to multiple disordered Zn interstitials that substantially reduce the phonon mean-free path[5]. Interstitial, disordered atomic positions are commonly observed in large complex crystal structures such as $Zn_4Sb_3$[5] and oxide-ion conductors[30], whereas they are rarely found in small simple crystal structures. The 1D interstitial, disordered, diffusive In$^{1+}$ ions discovered in InTe are quite noteworthy, given the simple structure with merely 8 atoms in primitive cell. The lattice thermal conductivity in both single-crystalline and polycrystalline InTe is unexpectedly low at high temperatures[16,23], which has been shown to reach the theoretical glass limit (~0.3 W m$^{-1}$ K$^{-1}$) proposed by Cahill[13] for the amorphous and disordered solids.

The interstitial, disordered, diffusive In$^{1+}$ ions could be considered as an important origin of the ultralow, weak temperature-dependent lattice thermal conductivity in InTe. The correlated disorder is known to lead to the broadening of phonon linewidth that is inversely proportional to lattice thermal conductivity[38]. Correlated disorder of indium vacancies and interstitials in 1D In$^{1+}$ chains could potentially broaden the phonon linewidth for the low-energy In$^{1+}$ vibration modes, resulting in the suppression of phonon lifetime, phonon mean-free path, and thereby lattice thermal conductivity. In particular, the structural disorder of In$^{1+}$ ions was suggested by Misra et al.[22] to be a key origin of the broadening of In$^{1+}$-weighted low-energy optical modes and the limited energy range of heat-carrying acoustic phonons, lowering

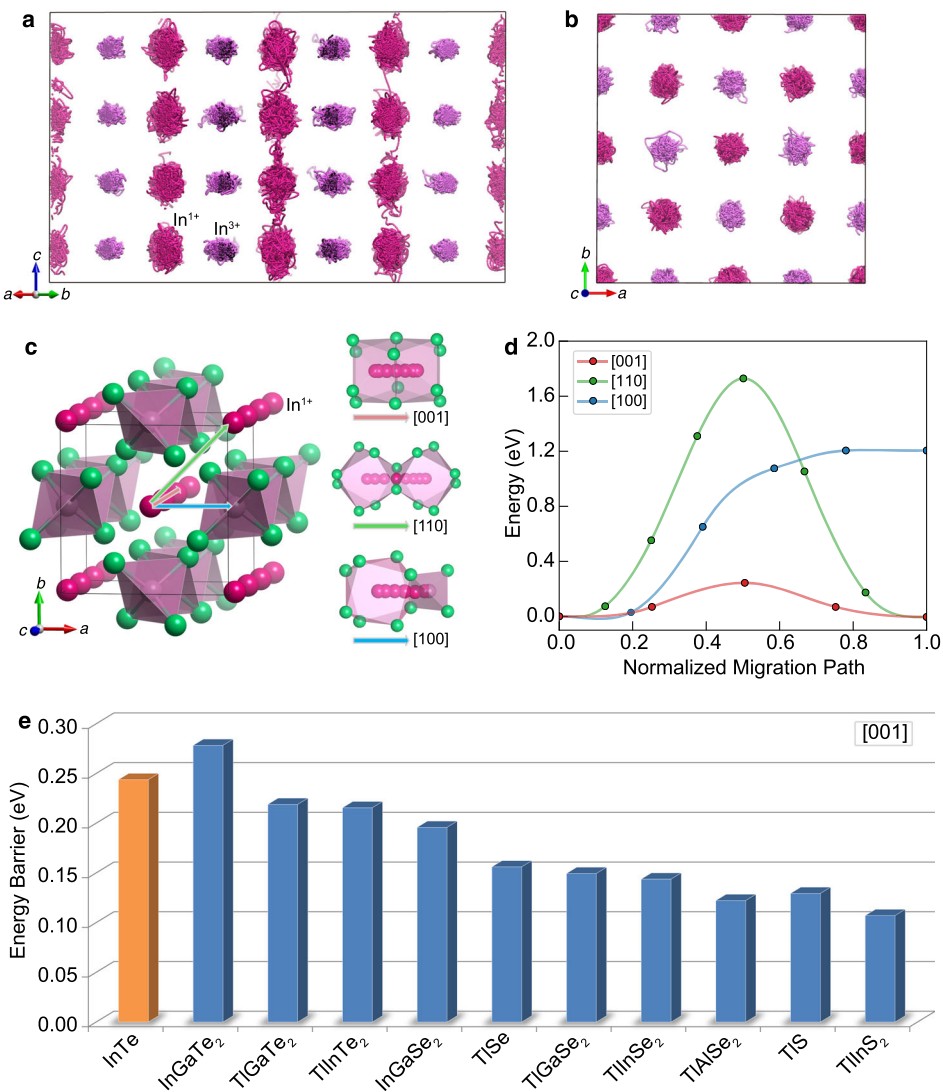

**Fig. 5 1D In$^{1+}$-ion migration pathways and energy barriers in InTe and several other ABX$_2$ compounds with the TlSe-type structure. a, b** Illustration of the 700 K molecular dynamics simulation trajectories of indium atoms in the In$^{1+}$-deficient supercell In$_{63}$Te$_{64}$ (In$_{0.984}$Te) projected along the [110] (**a**) and [001] (**b**) directions. **c, d** The vacancy-mediated In$^{1+}$-ion migration pathways (**c**) and calculated energy barriers (**d**) along the [001], [110], and [100] directions in InTe. **e** The activation energies of the vacancy-mediated A$^{1+}$-ion migration along the [001] direction in several ABX$_2$ compounds with the TlSe-type structure, in comparison with that in InTe.

the lattice thermal conductivity. Moreover, the plateau with nearly temperature-independent behavior ($\sim T^{-0.1}$) in thermal conductivity at ~25–80 K may be attributed to the correlated static disorder in 1D In$^{1+}$ chains revealed by the MEM density and 3D-$\Delta$PDF, similar to those observed in strongly disordered materials[39–41]. With increasing temperature above the Debye temperature of ~120 K, as the intrinsic phonon-phonon scattering begins to be dominant thermal conductivity shows a clear decreasing trend but its temperature dependence is still a bit weaker than $T^{-1}$, which is likely induced by the wavelike tunneling[11,14] contribution from interstitial, disordered, diffusive In$^{1+}$ ions. However, the exact impact of the structural disorder with interstitial, disordered, diffusive In$^{1+}$ ions comparing with the anharmonic phonon-phonon scattering on lattice thermal conductivity in InTe still requires further systematic theoretical and experimental investigations.

We expect the 1D disordered diffusion channel to be transferable to many other ABX$_2$ compounds with the TlSe-type structure. In Fig. 5e we show the calculated A$^{1+}$-ion migration energy barrier along the [001] direction in ten other ABX$_2$ compounds with the

TlSe-type structure in comparison with that in InTe. Notably, nearly all these compounds show comparable or lower migration energy barriers than that of InTe. This suggests that the 1D disordered diffusion channel of the A$^{1+}$-ions along the $c$ direction is very likely to appear in many other TlSe-type compounds, given commonly observed large thermal displacements and low migration energy barriers of the A$^{1+}$-ions along the $c$ direction. The intrinsic p-type transport behavior in nearly all other TlSe-type compounds may be understood by the dominant A$^{1+}$ cationic vacancy as described in this work. Considering the A$^{1+}$-ion diffusion has been widely mentioned in previous studies[19,20] without any experimental evidence to date, the direct observation of the 1D hopping/diffusion cationic channel in our work thereby provides long-awaited experimental evidence for understanding the electric-field enhanced superionic conductivity[19,20] observed in many TlSe-type compounds. Moreover, the proper description and fine details of the atomic disorder and hopping/diffusion in this work could be helpful for theory in developing models to understand ultralow thermal conductivity with weak temperature-dependent behavior in InTe and other TlSe-type chain-like materials.

## Methods

**Single crystal growth and characterization**. Large InTe single crystal was synthesized with the vertical Bridgman method. Indium shots (99.999%, ChemPUR) and tellurium pieces (99.999%, ChemPUR) were combined in stoichiometric ratio and placed into a carbon-coated quartz ampoule. The ampoule was flame-sealed after being evacuated to a pressure below $10^{-4}$ mbar. The sealed ampoule was then put into a box furnace, heated to 1023 K in 10 h with a dwell time of 24 h, and then slowly cooled to room temperature in 10 h. The obtained polycrystalline ingot was crushed into small pieces and loaded into a carbon-coated quartz ampoule with a conical tip, which was subsequently evacuated and sealed. For the crystal growth, the sample was kept at 1030 K for 24 h for homogenizing the melt and then slowly cooled directionally from the melt with a temperature gradient of ~6 K cm$^{-1}$ and a sample moving rate of 2 mm h$^{-1}$. To determine the c axis of the single crystal, X-ray diffraction data of cleavage surfaces were measured on a Rigaku Smartlab in the Bragg-Brentano geometry using a Cu Kα$_1$ source. The intersection line of the two perpendicular cleavage planes ($hh0$) defines the c axis. The chemical composition of the InTe single crystal was determined using the inductively coupled plasma optical emission spectrometry (ICP-OES). The sample was dissolved with PlasmaPure aqua regia and diluted to 1% acid concentration with MilliQ-water. The ICP-OES measurement was carried out on a Spectro ARCOS ICP-OES equipped with a Burgener Nebulizer and Cyclonic Spray Chamber with an ASX-520 Auto sampler. For quantification of the elements, the standard curves series was measured. The standard series consists of concentrations for each element with 0, 0.01, 0.05, 0.1, 0.5, 1, 5, 10, 20, and 50 ppm. The homogeneous elemental distribution of the InTe single crystal was confirmed with the Scanning Transmission Electron Microscopy Energy Dispersive Spectroscopy (STEM-EDS) by elemental mapping analysis using an FEI Talos F200X microscope operated at 200 kV. The microscope is equipped with an X-FEG electron source and Super-X EDS detector system. Small single crystals were extracted from the as-grown big crystal and screened with a Bruker Kappa Apex II diffractometer equipped with a Mo source ($\lambda = 0.71073$ Å). A small high-quality needle-shaped single crystal (~20 × 20 × 200 μm$^3$) was selected for the single-crystal synchrotron X-ray diffraction.

**Single-crystal synchrotron X-ray diffraction**. High-resolution single-crystal synchrotron X-ray diffraction data of InTe were collected at the BL02B1 beamline[24] from SPring-8 using a photon energy of 50.00 keV with a Pilatus3 X 1 M CdTe (P3) detector, which recently has been found capable of achieving extremely high quality for the electron density data[42]. The low-temperature data at 25, 100, 200, 300, and 400 K and the high-temperature data at 500, 600, and 700 K were collected with two different wavelengths of 0.2480 Å and 0.2509 Å on two different beam times, respectively. The collected frames were converted to the Bruker.sfrm format[42], which were integrated using SAINT-Plus[43]. After the integration, the data were scaled and corrected for absorption and other random errors using SADABS[44]. Subsequently, the data averaging and the uncertainty estimation were conducted using SORTAV[45]. The structure was solved with SHELXT[46], and the detailed structure refinement was conducted using JANA2006[47] (Supplementary Note 1 and Supplementary Tables 5–7). Details of the single-crystal diffraction data collection and reduction are summarized in Supplementary Table 3. The high-quality single-crystal synchrotron data here ensure accurate structure factors for obtaining high-quality, smooth MEM electron density.

**MEM density analysis**. Structure factors extracted from the structure refinement of high-resolution single-crystal synchrotron X-ray diffraction data were used for the calculations of MEM electron densities. The Sakato-Sato algorithm implemented in the BayMEM software[48] was used to conduct the MEM calculations. The unit cell of InTe was divided into $N_{pix} = 168 \times 168 \times 144$ pixels along the a, b, and c directions with a fine grid size of ~0.05 Å. Using scaled, phased, "error-free" structure factors as input, MEM calculations reconstruct and determine the electron density of the pixel i ($\rho_i$) that maximizes the information entropy, defined as

$$S = -\sum_{i=1}^{N_{pix}} \rho_i \ln\left(\frac{\rho_i}{\rho_i^{prior}}\right), \tag{1}$$

under the constraints based on the observed structure factors

$$\chi^2 = \frac{1}{N_F} \sum_{i=1}^{N_F} \left(\frac{|F_{obs} - F_{MEM}|}{\sigma(F_{obs})}\right)^2. \tag{2}$$

Here $N_F$ is the number of the observed structure factors, $F_{obs}$ is the observed experimental structure factors with standard uncertainties $\sigma(F_{obs})$, $F_{MEM}$ is the structure factors corresponding to the final calculated electron density, and $\rho_i^{prior}$ is the prior density that introduces prior information available for the system. Either a uniform prior density or a non-uniform prior density based on the independent spherical atom model may be used for the MEM calculations. In this study, the tests of MEM calculations with both a uniform (flat) prior and a nonuniform prior were conducted and the corresponding results were confirmed to be virtually identical (see Supplementary Fig. 4). For simplicity, the results of MEM calculations with uniform prior density were adopted for discussion in the main text. To ensure the optimal MEM density, the optimal stopping criterion $\chi^2$ of the MEM calculation was determined with the fractal dimension analysis of the residual density[49] (Supplementary Note 2 and Supplementary Figs. 9 and 10). The

MEM electron density is independent of different structural models, which is elucidated by virtually the same results using different structure models as input (Supplementary Fig. 3). For consistency, here we use the MEM electron density calculated with the structure factors from the full occupancy model for discussion in the main text. The 3D MEM electron density maps and crystal structures were visualized by VESTA[50].

**Diffuse scattering and 3D-ΔPDF**. In addition to the normal Bragg diffraction data for structural analysis and MEM calculations, X-ray diffuse scattering data of InTe single crystal at 25, 100, and 300 K were collected on the same setup at the BL02B1 beamline. Here 4 runs were measured, each a 180° Ω rotation with 900 frames, for χ = 0° and χ = 40°, with the detector at 2θ = 0° and 2θ = 20°. An exposure time of 2 s per frame was used. Background and air-scattering were measured using the same exposure time and detector positions as for the crystal. For each combination of these, 200 frames of air scattering were measured and averaged. The data were converted to reciprocal space using a custom Matlab script. During this process the data were corrected for Lorentz and polarization factors, the background scattering from air was subtracted, and a solid angle correction was applied as the detector is flat. The resulting scattering data were reconstructed on a 901 × 901 × 901 point grid with each axis spanning ± 28.3 Å$^{-1}$. The resulting data were symmetrized using the $4/mmm$ point symmetry of the Laue group, and outlier rejection was used in the symmetrization, such that symmetry equivalent voxels were compared and rejected if they deviated by more than two standard deviations from the median of the equivalents[45]. The Bragg peaks were punched and filled. Because a large degree of diffuse scattering is observed at the positions of the Bragg peaks, care has to be taken to remove all Bragg scattering while leaving as much of the diffuse scattering as possible. To accomplish this, the Bragg peaks were masked in the raw data frames before conversion using a Python script which masks sharp and strong peaks in the data. This is possible as the diffuse scattering here is slowly-varying compared to Bragg peaks. The script compares the measured pixels to all neighboring pixels, and if a pixel is relatively much stronger (a factor of 2.5 was used here) than its neighbors, it is marked as part of a Bragg peak. To avoid weak noise from being marked as peaks, a lower bound for the absolute intensity needed for a peak is also used. A high-intensity cutoff is also used, for which pixels larger than this value are automatically marked as peaks. Once peak pixels have been marked this way, the pixels neighboring these are also marked as being part of peaks. This approach effectively masks most Bragg peaks, but leaves some of the almost-zero-intensity Bragg peaks at large scattering vectors. To fill in the masked regions and remove remaining weak peaks, a small punch was applied to the allowed Bragg positions after conversion to reciprocal space and linear interpolation was used to fill. A constant value was also filled into the regions where no data has been measured to minimize Fourier ripples. The 3D-ΔPDF is obtained as the inverse Fourier transform of the diffuse scattering intensity, $I_d$, and is given by the autocorrelation of the deviations in electron density from the average crystal structure[34]:

$$\text{3D-}\Delta\text{PDF} = \mathcal{F}^{-1}[I_d] = \langle \delta\rho \otimes \delta\rho \rangle. \tag{3}$$

Here $\delta\rho(\mathbf{r}, t) = \rho(\mathbf{r}, t) - \rho_{periodic}(\mathbf{r})$ is the difference between the total electron density of the crystal and the periodic average electron density. The brackets $\langle \dots \rangle$ indicate time averaging, $\otimes$ denotes cross correlation, and $\mathcal{F}$ is the Fourier transform.

**Thermal conductivity measurements**. A small bar-shaped sample obtained by cutting the large single crystal along the c axis was used for low-temperature thermal conductivity measurement. Low-temperature thermal conductivities from 2 to 300 K were measured with a Physical Property Measurement System (PPMS, Quantum Design, US) using the Thermal Transport Option (TTO) under high vacuum. A flat sample (~ 6 × 6 × 1.9 mm$^3$) with the surface normal along the c axis was extracted from the large single crystal and used for high-temperature thermal diffusivity measurement. The thermal diffusivity ($D$) measurement from 300 to 723 K was carried out on a Netzsch LFA457 setup using the laser flash method. High-temperature thermal conductivity was then determined using $\kappa = dDC_P$, where the heat capacity $C_P$ was estimated using the Dulong-Petit law $C_P = 3k_B$ per atom and the sample density $d$ was measured with the Archimedes method.

**Theoretical calculations**. All density functional theory calculations were conducted with the Perdew-Burke-Ernzerhof (PBE) generalized gradient approximation[51] based on the projector-augmented wave method[52] (PAW) as implemented in the Vienna ab initio simulation package[53] (VASP). Defect calculations were conducted in a supercell of 2 × 2 × 2 conventional unit cells. A Γ-centered 4 × 4 × 4 k-point mesh was used for the structure optimization as well as the total energy calculation. A plane wave cutoff energy of 400 eV and an energy convergence criterion of $10^{-4}$ eV were applied. For the structural optimization of defect supercells, all atomic positions were relaxed into their equilibrium positions until the Hellmann-Feynman force was converged to be smaller than 0.01 eV Å$^{-1}$ while the lattice parameters were fixed at the optimized values from the perfect supercell. For the defect supercells with interstitial indiums (In$_i$(1) and In$_i$(2)), the positions of interstitial indiums were fixed at the experimental values while the atomic positions of other atoms were fully relaxed. The formation energy of a

native point defect was simply calculated using[54]:

$$\Delta E_f^d = E_{tot}^d - E_{tot}^{bulk} - \sum_i n_i \mu_i, \qquad (4)$$

where $\Delta E_f^d$ is the total energy of the defect supercell, $E_{tot}^{bulk}$ is the total energy of the bulk supercell without any defects, $n_i$ denotes the number of atoms of type $i$ that is added to ($n_i > 0$) or removed from ($n_i < 0$) the bulk supercell, and $\mu_i$ represents the atomic chemical potential. The chemical potentials of In and Te in InTe should fulfill the thermodynamic condition: $\Delta\mu_{In} + \Delta\mu_{Te} = 2\Delta H_f(InTe)$. Under the In-rich condition, the chemical potentials are limited by the formation energy of the secondary phase $In_4Te_3$ ($4\Delta\mu_{In} + 3\Delta\mu_{Te} \leq 7\Delta H_f(In_4Te_3)$). Under the In-poor condition, the chemical potentials are limited by the formation energy of the competing phase $In_3Te_4$ ($3\Delta\mu_{In} + 4\Delta\mu_{Te} \leq 7\Delta H_f(In_3Te_4)$).

The calculations of minimum-energy diffusion pathways and migration barriers of $In^{1+}$ ions in InTe were performed with the climbing image nudged elastic band (CI-NEB) method[55]. The calculations were conducted in a $2 \times 2 \times 2$ supercell of the conventional unit cell, where one vacancy was introduced on the $In^{1+}$ sites. We calculated the diffusion energy barriers for a single nearest indium ion diffusing to the created vacancy. The atomic positions were relaxed until the residual forces were within 10 meV Å$^{-1}$. A total energy convergence criterion of $10^{-6}$ eV and a $\Gamma$-centered $2 \times 2 \times 2$ k-point mesh were used for the CI-NEB calculations. The same calculation procedure was applied to calculate the migration barriers of the $A^{1+}$ ion along the [001] direction in several other $ABX_2$ compounds (including $InGaTe_2$, $TlGaTe_2$, $TlInTe_2$, $InGaSe_2$, $TlAlSe_2$, $TlGaSe_2$, $TlInSe_2$, $TlInS_2$, TlSe, and TlS) with the TlSe-type structure.

Ab initio molecular dynamics (MD) calculations were performed using the PBE functional as implemented in the VASP code. The MD simulations at 700 K were conducted with the NVT ensemble using a Nosé thermostat with the default Nosé mass. A supercell with 127 atoms ($2 \times 2 \times 2$ conventional cells) containing one $In^{1+}$ vacancy was used so that the simulated composition $In_{63}Te_{64}$ ($In_{0.984}Te$) is close to the experimental observation. A 300 eV cutoff energy and an energy convergence criterion of $10^{-5}$ eV along with the $\Gamma$-point sampling were adopted for the simulations. The simulations were run for 130 ps for statistical analysis with a time step of 2 fs, and the first 10 ps was disregarded for equilibrium. The trajectories of the MD simulations were visualized by VMD[56].

## Data availability
The data supporting the findings of this study are available within the paper and its Supplementary Information files and are available from the corresponding authors upon reasonable request.

## Code availability
Custom computer codes used in this work are available from the corresponding authors upon reasonable request.

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

## Acknowledgements

Aref Mamakhel is acknowledged for the STEM-EDS measurement. Mads R. V. Jørgensen is thanked for fruitful discussions. This work was supported by the Villum Foundation and the Danish Agency for Science, Technology and Innovation (DanScatt). Affiliation with the Aarhus University Center for Integrated Materials Research (iMAT) is gratefully acknowledged. This work was partly supported by the Japan Society for the Promotion of Science (JSPS) Grants-in-Aid for Scientific Research (KAKENHI) Grant Number JP19KK0132 and JP20H4656. The synchrotron experiments were performed at SPring-8 BL02B1 with the approval of the Japan Synchrotron Radiation Research Institute (JASRI) as a Partner User (Proposals No. 2017B0078, No. 2018A0078, No. 2018B0078, and No. 2019A0159). Beamline scientist K. Sugimoto is acknowledged for support during syn- chrotron experiments. The theoretical calculations in this work were conducted at the Center for Scientific Computing in Aarhus (CSCAA).

## Author contributions

J.Z. designed the project under the supervision of B.B.I. J.Z. prepared and characterized the single crystals. J.Z. carried out the data analysis, structure refinement, and MEM density analysis. N.R. analyzed the diffuse scattering data and 3D-ΔPDF. K.T. provided guidance on the analysis of synchrotron data and MEM density. S.T. and E.N. collected single-crystal synchrotron X-ray diffraction data. J.Z. conducted theoretical calculations. J.Z. and L.S. conducted thermal conductivity measurements. M.B. conducted the ICP-OES measure-ment. J.Z., N.R., K.T., L.S., and B.B.I. contributed to the discussion of the results. J.Z., N.R., and B.B.I. wrote the manuscript. All other authors read and edited the manuscript.

## Competing interests

The authors declare no competing interests.
