## [Peer Review File · Nature Communications]

Editorial Note: Parts of this peer review file have been redacted as indicated to remove third-party material where no permission to publish could be obtained.

REVIEWER COMMENTS

Reviewer #1 (Remarks to the Author):

In this manuscript titled as "Direct observation of one-dimensional disordered diffusion channel in a chain-like thermoelectric with ultralow thermal conductivity", the authors have investigated the structure of InTe and its correlation with low thermal conductivity. Using a combination of SXRD, MEM analysis and 3D- Δ PDF, they have found that InTe crystallizes as $\text{In}_{0.98}\text{Te}$, which in essence makes it an electron deficient p-type semiconductor. The structural analysis concluded that there are two interstitial In^+ sites forming a 1D disordered chain along the c-axis. Temperature dependent 3D- Δ PDF provided conclusive direct evidence of this diffusion pathway, which explains its superionic and low thermal conductive behavior. I feel the work here is of high novelty and explained in a very detailed way. I recommend the manuscript to be published in Nature Communications pending few of my queries (minor revision).

1. The authors did not explain how these interstitial In^+ help in reducing the phonon propagation through the system. The correlation between this structural disorder with thermal conductivity should be presented more clearly.
2. In Figure 4a and b, there seems small satellite peaks away from the x-direction. Does that mean that small amounts interstitial In^+ ions are also placed along the x-axis?
3. Did the authors refine the occupancy of In^{3+} and Te as well? If so, what's their error range?
4. In the 3D- Δ PDF of InTe, only the correlation between two $\text{In}(i)$ atoms at 5.8 Å are shown. Why there are no signatures of other $\text{In}(i)$ - $\text{In}(i)$ correlations present?

The article addresses an interesting topic that combining experimental (SSCXRD) methods and statistical algorithm (MEM) to verify the existence of one-dimensional disordered diffusion channel in InTe single crystal which is regarded as one cause of ultralow thermal conductivity of InTe single crystal or other TlSe-type chain-like materials by authors. The article is generally good, but there are some issues which need to be fixed before its suitability for publication can be assessed further.

1. Line 48: Except one-dimensional disordered diffusion channel validated by the article, authors also mention that ultralow thermal conductivity of InTe is typically attributed to strong anharmonic rattling. The author should further explain to what extent the ultralow thermal conductivity of InTe is attributed to strong anharmonic rattling or two interstitial indium sites. If ultralow thermal conductivity of InTe is almost entirely caused by strong anharmonic rattling, the discussion of the article is meaningless.
2. Fig. 2a and Fig. 3a are the electron density maps plotted by MEM calculations through using SSCXRD data, how can author call them “direct observation”? Even though the experimental method of SSCXRD is carried on, there are still assumptions and speculations especially in MEM during verifying the existence of two interstitial indium sites in InTe crystal. My understanding is that the validation of disordered diffusion channel in InTe crystal presented by this article is more suitable referring as “theoretical calculation” rather than “direct observation”.
3. Line 56: Authors use SSCXRD as abbreviation of “synchrotron single-crystal X-ray diffraction” firstly but later change to SCSXRD.
4. Line 94: How to define “uniform distribution” of In and Te from the STEM picture.
5. Line 103: Can author explain more about changes of decreasing trend of thermal conductivity of InTe single crystal with temperature? Why first $K \sim T^{-0.1}$ and then $K \sim T^{-0.69}$?

Reviewer #3 (Remarks to the Author):

In this manuscript, the authors report on a detailed experimental study of the structural disorder characterizing the chalcogenide semiconductor InTe. This compound has recently attracted attention due to its good thermoelectric performance and its surprisingly very low lattice thermal conductivity. Using a combination of single-crystal diffraction techniques and calculations, they notably show that the In⁺ cations that reside in large tunnels of the structure are significantly disordered, which is thought of as being the main feature driving the lattice thermal conductivity to the very low values measured experimentally. The growth of single-crystalline InTe has already been reported in the literature together with a detailed investigation of its transport properties and lattice dynamics based on inelastic neutron scattering. In prior studies on InTe, the presence of vacancies, the disordered nature of the In⁺ cations and their role on the thermal transport have also been thoroughly discussed. This leaves the experimental validation of the disordered nature of the In⁺ cations the only novelty reported in the manuscript. For these reasons, although well performed, this study falls short of the level of novelty and impact required to be publishable in Nature Communications. This manuscript will be a better fit for a more specialized journal such as Scientific Report. Prior to resubmitting, several important points, listed below, must be addressed to make the manuscript clearer and more concise, notably regarding the connection between the above-mentioned crystallographic feature and the thermal transport.

1). Several times throughout the manuscript, the authors use the term “diffusion” to characterize the In⁺ cations. It is however not clear whether this phenomenon remains localized in the cages formed by the octahedral chains or whether a truly ionic conduction takes place. According to prior studies, this mechanism has not been observed. This point should be clarified in the manuscript. In addition, it would be worthwhile to recall that ionic conductors cannot be considered as serious candidates for integration in thermoelectric generators due to the rapid deterioration of their performance this mechanism implies.

2). A comparison of the temperature dependences of the lattice thermal conductivity (Figure 1d) and of the electrical resistivity (Figure in the SI) with those reported previously (Ref. 22 in the manuscript) would be interesting. In particular, the dielectric maximum in the lattice thermal conductivity is observed near 4 K in the present study while it seems it was not yet reached at 2 K in Ref. 22. This may point to differences in crystalline quality, perhaps due to different vacancy concentration. To clarify this point, the hole concentration of the present crystal should be measured and discussed. This would allow a connection with the chemical composition reported (In_{0.98}Te) to be possibly made. More precisely, is the hole concentration consistent with this composition assuming one hole per In⁺ vacancy? Moreover, in Ref. 16, much lower In vacancy concentrations have been suggested in polycrystalline samples. A detailed discussion about this point should be provided in the manuscript. Finally, the authors should also mention that the values of the thermal conductivity at high temperatures are similar in both crystallographic directions (hh0 plane and c direction) according to the results reported in Ref. 30, showing that the thermal transport in single-crystalline InTe is nearly isotropic.

3). The kink observed in the electrical resistivity curve has been already observed and discussed in prior studies (Ref. 22 and S. Y. Back, ACS Applied Energy Mater. 3, 3628 (2020), this last study is not mentioned in the manuscript). The authors should mention that this kink is consistently observed in both single-crystalline and polycrystalline InTe at low temperatures and is thus not due to a measurement artifact. In addition, the large thermal displacement parameters of the In⁺ cations had been originally observed in the study of T. Chattopadhyay et al. (J. Phys. Chem. Solids, 46, 351 (1985)). This should be clearly mentioned in the present manuscript as well.

4). The authors mention in their manuscript that “lone-pair-induced strong anharmonicity, the interstitial, disordered, diffusive In¹⁺ ions thereby should be considered as a key origin of the ultralow thermal conductivity as well as its weak temperature dependence in InTe.” This sentence is a bit

misleading for readers since the link between the disordered In⁺ cations and the lattice thermal conductivity has already been considered as indeed being a key origin in Ref. 22. This study has notably showed how this structural characteristic gives rise to low-energy optical modes with a low dynamic structure factor, as observed by inelastic neutron scattering. A more detailed discussion based on these results should be provided in the present manuscript.

Reviewer 1

Comment 1

'In this manuscript titled as "Direct observation of one-dimensional disordered diffusion channel in a chain-like thermoelectric with ultralow thermal conductivity", the authors have investigated the structure of InTe and its correlation with low thermal conductivity. Using a combination of SXRD, MEM analysis and 3D- Δ PDF, they have found that InTe crystallizes as In_{0.98}Te, which in essence makes it an electron deficient p-type semiconductor. The structural analysis concluded that there are two interstitial In⁺ sites forming a 1D disordered chain along the c-axis. Temperature dependent 3D- Δ PDF provided conclusive direct evidence of this diffusion pathway, which explains its superionic and low thermal conductive behavior. I feel the work here is of high novelty and explained in a very detailed way. I recommend the manuscript to be published in Nature Communications pending few of my queries (minor revision).'

Reply

We thank the referee for his/her careful reading and recommendation for publication.

Comment 2

'1. The authors did not explain how these interstitial In⁺ help in reducing the phonon propagation through the system. The correlation between this

structural disorder with thermal conductivity should be presented more clearly.'

Reply

Thank you for the comment. In the revised manuscript, we have done corresponding revisions to explain how disordered In^{1+} ions help in reducing thermal conductivity (see below as well as changes in the revised manuscript). A discussion on the correlation between the structural disorder and thermal conductivity is added to make a more clear presentation. Since the focus of the current work is on a systematic study of structural disorder in InTe and the current paper is already quite long, we do not show more detailed discussion or investigation given the constraints imposed by journal style. We hope that the current systematic structural disorder study will provide a basis for a future theoretical study on the correlation between this structural disorder and thermal conductivity.

To address this comment, the following revision is applied:

One new paragraph is added in the Discussion section on page 10: “The interstitial, disordered, diffusive In^{1+} ions should be considered as an important origin of the ultralow, weak temperature dependent lattice thermal conductivity in InTe. Correlated disorder is known to lead to the broadening of phonon linewidth that is inversely proportional to lattice thermal conductivity³⁸. Correlated disorder of indium vacancies and interstitials in 1D In^{1+} chains could potentially broaden the phonon linewidth for the low-energy In^{1+} vibration modes, resulting in the suppression of phonon lifetime, phonon mean-free path, and thereby lattice thermal conductivity. In particular, the structural disorder of In^{1+} ions was suggested by Misra et al.²² to be a key origin of the broadening of In^{1+} -weighted low-energy optical modes and the limited energy range of heat-carrying acoustic phonons, lowering the lattice thermal conductivity. Moreover, the plateau with nearly temperature-independent behavior ($\sim T^{-0.1}$) in thermal conductivity at ~ 25 -80 K may be attributed to the correlated static disorder in 1D In^{1+} chains revealed by the MEM density and 3D- Δ PDF, similar to those observed in strongly disordered materials³⁹⁻⁴¹. With increasing temperature above the Debye temperature of ~ 120 K, as the intrinsic phonon-phonon scattering begins to be dominant thermal conductivity shows a clear decreasing trend but its temperature dependence is still a bit weaker than T^{-1} , which is likely induced by the wavelike tunneling^{11,14} contribution from interstitial, disordered, diffusive In^{1+} ions. However, the exact impact of the structural disorder with interstitial, disordered, diffusive In^{1+} ions comparing with the anharmonic phonon-phonon scattering on lattice thermal conductivity in InTe still requires further systematic theoretical and experimental investigations.”

Several new references (refs. 38-41) regarding the thermal transport in disordered materials have been added.

Comment 3

'2. In Figure 4a and b, there seems small satellite peaks away from the x-direction. Does that mean that small amounts interstitial In⁺ ions are also placed along the x-axis?'

Reply

Thank you for the comment. In Figure 4a there is diffuse scattering also for non-integer values of L . This means that there are also correlations in the structure that have non-zero x components. This is also seen in Figure 4b, where features are clearly seen in the 3D- Δ PDF away from the $x=0$ axis. However, the vectors for these features in the 3D-delta-PDF do not correspond to any interstitial sites, but just to vectors between the ideal sites in the structure. The shape of these features (negative-positive-negative) in the direction to the origin are typical indications of simple in-phase motion of atoms corresponding to acoustic phonons close to the gamma point.

Comment 4

'3. Did the authors refine the occupancy of In³⁺ and Te as well? If so, what's their error range?'

Reply

Thanks for the comment. We have refined the occupancy of In³⁺ and Te, and generally got the values very close to 1 (e.g. 1.0008 ± 0.0015 and 0.9995 ± 0.0014 respectively for In³⁺ and Te at 25 K with the two interstitial model). As the In³⁺ and Te are generally fully occupied, to reduce the refinement parameters for simplicity we fix the occupancy of In³⁺ and Te to unity and only refine the occupancy of In¹⁺. The corresponding revision is summarized below:

One sentence is added in Supplementary Note 1: "...As the refined site occupancies for In³⁺ and Te are generally very close to unity (i.e. 1.0008(15) and 0.9995(14) respectively for In³⁺ and Te at 25 K), we therefore only refined the site occupancy of In¹⁺ and indium interstitials...."

Comment 5

'4. In the 3D- Δ PDF of InTe, only the correlation between two In(i) atoms at 5.8 Å are shown. Why there are no signatures of other In(i)-In(i) correlations present?'

Reply

Thanks for the comment. The 3D- Δ PDF shows the correlations compared to a random distribution (uncorrelated). That strong positive correlations are not seen for other possible In(i)-In(i) vectors indicate that indium interstitials tend not to be separated by those other vectors in the real structure. For many of the other possible vectors along the x direction there is a weak negative signal, suggesting In interstitials tend not to be separated by those vectors. When looking at vectors between interstitials in different chains, no significant signal is observed, suggesting they are uncorrelated.

Reviewer 2

Comment 1

'The article addresses an interesting topic that combining experimental (SSCXRD) methods and statistical algorithm (MEM) to verify the existence of one-dimensional disordered diffusion channel in InTe single crystal which is regarded as one cause of ultralow thermal conductivity of InTe single crystal or other TlSe-type chain-like materials by authors. The article is generally good, but there are some issues which need to be fixed before its suitability for publication can be assessed further'

Reply

We thank the referee for his/her careful reading and comments. Below we address the comments in detail.

Comment 2

'1. Line 48: Except one-dimensional disordered diffusion channel validated by the article, authors also mention that ultralow thermal conductivity of InTe is typically attributed to strong anharmonic rattling. The author should further explain to what extent the ultralow thermal conductivity of InTe is attributed to strong anharmonic rattling or two interstitial indium sites. If ultralow thermal conductivity of InTe is almost entirely caused by strong anharmonic rattling, the discussion of the article is meaningless.'

Reply

Thank you for the comment. In the revised manuscript, we have done corresponding revisions to explain how disordered In^{1+} ions help in reducing thermal conductivity. See the reply to comment 2 of the first reviewer for details.

The structural disorder typically breaks local symmetry and periodicity, making the conventional phonon-gas model with anharmonic phonon-phonon scattering no longer applicable. Lattice thermal conductivity of InTe has been shown to reach the theoretical glass limit ($\sim 0.3 \text{ W m}^{-1} \text{ K}^{-1}$) for the amorphous and disordered solids, a clear indication of the important impact of the disordered In^{1+} ions on lattice thermal conductivity. Correlated disorder of indium vacancies and interstitials in 1D In^{1+} chains could potentially broaden the phonon linewidth for the low-energy In^{1+} vibration modes, resulting in the suppression of phonon lifetime, phonon mean-free path, and thereby lattice thermal conductivity. In particular, the structural disorder of In^{1+} ions was suggested by Misra et al. (*Phys. Rev. Res.* **2**, 043371 (2020)) using inelastic neutron scattering to be a key origin of the broadening of In^{1+} -weighted low-energy optical modes and the limited energy range of heat-carrying acoustic phonons, lowering the lattice thermal conductivity. Moreover, the abnormal weak temperature dependent thermal conductivity below Debye temperature is a clear impact of static disorder (see the reply to comment 6 for details). All these reveal that the impact of structural disorder should not be overlooked and should be considered as an important origin for ultralow, weak temperature dependent thermal conductivity in InTe.

It still remains a great challenge for current theory to accurately model structural disorder and its quantitative impact on thermal transport. A systematic theoretical model (e.g. supercell lattice dynamics, two-channel model, or unified theory) considering the structural disorder shown in this work needs to be conducted in future to understand more quantitatively the impact of the structural disorder on ultralow thermal conductivity in InTe. As the focus of our paper is on the systematic investigation of structural disorder and the length and content of our paper is already very substantial, we feel that the detailed investigation of the exact impact of structural disorder using a comprehensive theoretical model should be conducted and presented in a future project.

Comment 3

'2. Fig. 2a and Fig. 3a are the electron density maps plotted by MEM calculations through using SCSXRD data, how can author call them "direct observation"? Even though the experimental method of SCSXRD is carried on, there are still assumptions and speculations especially in MEM during

verifying the existence of two interstitial indium sites in InTe crystal. My understanding is that the validation of disordered diffusion channel in In-Te crystal presented by this article is more suitable referring as “theoretical calculation” rather than “direct observation”.’

Reply

Thank you for the comment. We are sorry for making you confused about this point. Below we explain in detail why it is “direct observation”.

The maximum entropy method (MEM) is not “theoretical calculation” but an analysis method for experimental data based on Bayesian statistics. MEM is an information-theory-based technique and an early example is by Gull and Daniel (*Nature* **272**, 686-690 (1978)) in the field of radioastronomy to enhance the information from noisy data. Collins (*Nature* **298**, 49-51 (1982)) applied the methodology to crystallography for electron density enhancement from X-ray diffraction. The MEM enables us to extract the maximum amount of experimental information from observed diffraction data with least bias. Successful MEM enhancement makes it possible to evaluate not only the missing and heavily overlapped reflections but also any type of complicated electron or nuclear distribution, which is difficult to describe with classical structural model. In our group we have during the past three decades published a long list of structural studies using the MEM starting from the discovery of non nuclear electron density maxima in 1995 (*Acta Crystallogra Sect B* **51**, 580 (1995))

It should be noted that the present MEM density is independent of structure models or prior densities. This is shown with the identical 1D MEM density profile with different structure models and prior densities (see Supplementary Figs. 3 and 4, also shown below). This clearly reveals that the experimental diffraction data strongly constrains the MEM electron density and it is therefore very reasonable to use “direct observation”.

Supplementary Fig. 3 | Comparison of 1D MEM electron density profiles with different structure models. The test of MEM calculations at 25 K is based on the structure factors extracted with the flat prior density as well as $\chi^2 = 0.02$.

Supplementary Fig. 4 | Comparison of 1D MEM electron density profiles with flat prior and non-uniform prior density. The test of MEM calculations at 25 K is based on the structure factors extracted with the full occupancy model as well as $\chi^2 = 0.02$.

A simple google search with “direct observation” and “maximum entropy method” results in more than hundred records with published papers. To further address the reviewer’s concern, below we also list three published examples that use the MEM technique for the “direct observation” of the electron/nuclear density distributions.

a) In *Science* **298**, 2358-2361 (2002): *Kitaura et al.* reported the “direct observation” of dioxygen molecules physisorbed in the nanochannels of a microporous copper coordination polymer by the MEM (maximum entropy method)/Rietveld method, using in situ high-resolution synchrotron x-ray powder diffraction measurements.

b) In *Nat. Mater.* **3**, 458-463 (2004): *Snyder et al.* reported the “experimental observation” of disordered zinc in Zn₄Sb₃ using the MEM electron density for understanding phonon-glass and electron-crystal thermoelectric properties.

c) In *Nat. Mater.* **7**, 707-711 (2008): *Nishimura et al.* reported the “experimental visualization” of lithium diffusion in Li_xFePO₄ using the MEM technique.

Comment 4

'3. Line 56: Authors use SSCXRD as abbreviation of "synchrotron single-crystal X-ray diffraction" firstly but later change to SCSXRD.'

Reply

Thanks very much for pointing out the mistake. In the revised manuscript, we have revised and ensured the same abbreviation (SCSXRD: *single-crystal synchrotron X-ray diffraction*) consistently throughout the paper.

Comment 5

'4. Line 94: How to define "uniform distribution" of In and Te from the STEM picture.'

Reply

Thanks for the comment. In Supplementary Fig. 1, we show the STEM-EDS elemental mapping of a small InTe crystal. The elemental maps and the overlay of In and Te elements reveal generally a uniform distribution of In and Te in the InTe crystal.

Supplementary Fig. 1 | STEM-EDS elemental mapping of a small crystal extracted from the as-grown large InTe single crystal.

Comment 6

'5. Line 103: Can author explain more about changes of decreasing trend of thermal conductivity of InTe single crystal with temperature? Why first $K \sim T^{-0.1}$ and then $K \sim T^{-0.69}$?'

Reply

Thanks for the comment. In most single-crystalline materials, the thermal conductivity generally shows a crystalline peak at low temperatures, and then κ monotonically decreases with increasing temperature above Debye temperature with a typical phonon-phonon scattering temperature dependence of T^{-1} . In InTe single crystals, the thermal conductivity shows a plateau with nearly temperature-independent behavior ($\sim T^{-0.1}$) at ~ 25 -80 K, where the extrinsic scattering such as structural disorder/defect is more important than intrinsic phonon-phonon scattering since the temperature is below the Debye temperature of ~ 120 K. Such glass-like plateau commonly observed in strongly disordered materials is mainly caused by the static disorder (e.g. *Cryst. Res. Technol.* **52**, 1700114 (2017); *Phys. Rev. B* **4**, 2029-2041 (1971); *J. Appl. Phys.* **119**, 185102 (2016); etc.). The plateau with nearly temperature-independent behavior ($\sim T^{-0.1}$) in InTe thereby could be attributed to the correlated static disorder in 1D In^{1+} chains revealed by the MEM density and 3D- Δ PDF. With increasing temperature above the Debye temperature of ~ 120 K, as the intrinsic phonon-phonon scattering begins to be dominant the thermal conductivity shows a clear decreasing trend but its temperature dependence is still a bit weaker than T^{-1} ($T^{-0.69}$), which may be attributed to the wavelike tunneling (*Science* **360**, 1455-1458 (2018); *Nat. Phys.* **15**, 809-813 (2019)) contribution from interstitial, disordered, diffusive In^{1+} ions.

To address this comment, a few sentences are added in the Discussion section of the revised manuscript: “Moreover, the plateau with nearly temperature-independent behavior ($\sim T^{-0.1}$) in thermal conductivity at ~ 25 -80 K may be attributed to the correlated static disorder in 1D In^{1+} chains revealed by the MEM density and 3D- Δ PDF, similar to those observed in strongly disordered materials³⁹⁻⁴¹. With increasing temperature above the Debye temperature of ~ 120 K, as the intrinsic phonon-phonon scattering begins to be dominant thermal conductivity shows a clear decreasing trend but its temperature dependence is still a bit weaker than T^{-1} , which is likely induced by the wavelike tunneling^{11,14} contribution from interstitial, disordered, diffusive In^{1+} ions.”

Reviewer 3

Comment 1

‘In this manuscript, the authors report on a detailed experimental study of the structural disorder characterizing the chalcogenide semiconductor InTe. This compound has recently attracted attention due to its good thermoelectric performance and its surprisingly very low lattice thermal con-

ductivity. Using a combination of single-crystal diffraction techniques and calculations, they notably show that the In⁺ cations that reside in large tunnels of the structure are significantly disordered, which is thought of as being the main feature driving the lattice thermal conductivity to the very low values measured experimentally. The growth of single-crystalline InTe has already been reported in the literature together with a detailed investigation of its transport properties and lattice dynamics based on inelastic neutron scattering. In prior studies on InTe, the presence of vacancies, the disordered nature of the In⁺ cations and their role on the thermal transport have also been thoroughly discussed. This leaves the experimental validation of the disordered nature of the In⁺ cations the only novelty reported in the manuscript. For these reasons, although well performed, this study falls short of the level of novelty and impact required to be publishable in Nature Communications. This manuscript will be a better fit for a more specialized journal such as Scientific Report. Prior to resubmitting, several important points, listed below, must be addressed to make the manuscript clearer and more concise, notably regarding the connection between the above-mentioned crystallographic feature and the thermal transport.'

Reply

We thank the referee for the comments, but we clearly disagree with the comment about the novelty of the paper. Below we explain in detail the novelty of the scientific content of our work.

- a) Although the previous study has suggested that many interesting physical properties might be induced by the disordered nature of the In¹⁺ cations, all these are based on the hypothesis without any direct evidence. The authors in Ref. 22 also clearly noted in their paper that the presence of local disorder induced by the In¹⁺ site occupancy is a hypothesis. In fact, the structural disorder and ion diffusion have been suggested many times in many early studies as the possible origin of the electric-field driven superionic conductivity in many TlSe-type compounds (e.g. *Jpn. J. Appl. Phys* **50**, 05FC09 (2011); *Semiconductors* **45**, 1387-1390 (2011)). However, no direct experimental evidence of the structural disorder or ion diffusion has been reported so far in any TlSe-type compounds. This has become a long-standing scientific question not only for InTe but more generally for TlSe-type materials since without direct experimental evidence, a clear understanding of the effect of structural disorder on many physical properties cannot be rationalized. A systematic study of structural disorder is thus of great importance for understanding physical properties in TlSe-type compounds in general. The present study for the first time reports experimental data confirming the disorder.

- b) Significant theoretical models such as the two-channel model (*Science* 2018, 360, 1455-1458) and unified theory (*Nat. Phys.* 2019, 15, 809-813) recently developed have been quite successful in modeling ultralow, weak-temperature-dependent thermal conductivities of crystalline solids through introducing a wave-like tunneling term for describing the disorder. However, detailed experimental evidence on the disorder is generally lacking in many simple crystalline solids. This is typically because of great challenges for experimentalists to probe subtle structural disorders in simple crystalline solids. Advanced crystallographic methods such as the maximum entropy method (MEM) and 3D- Δ PDF require very high quality diffraction data, and for the latter techniques this must be done on high quality single crystals. These are very challenging tasks and in the present case crystal growth, data collection and analysis indeed has taken three years. Therefore, we do not think that our systematic structural disorder study falls short of the level of novelty and impact required to be publishable in Nature Communications.
- c) A systematic study of structural disorder by the MEM and 3D- Δ PDF does not fall short of the level of novelty and impact. Numerous structural studies using merely the MEM technique have been published in high-impact journals including Nature, Science, Nature Materials, Nature Communications, etc. Lots of these studies also simply provided important experimental evidence of missing atoms, structural disorder, or ionic diffusion pathways, which have been previously suggested by either unusual properties or theoretical predictions with no experimental evidence. This clearly documents that a structural study alone should not be considered as short of the level of novelty and impact. Here we show several high-impact examples highlighting the importance of structural analysis with the MEM technique: 1) Lithium iron phosphate was discovered as a new class of cathode materials in 1997. However, lithium diffusion was generally limited to computational predictions without experimental evidence. In *Nat. Mater.* **7**, 707-711 (2008): *Nishimura et al.* was able to report the “experimental visualization” of lithium diffusion in Li_xFePO_4 thanks to the MEM technique. 2) In *Nat. Mater.* **3**, 458-463 (2004): *Snyder et al.* reported the experimental evidence of disordered zinc in Zn_4Sb_3 using the MEM electron density for understanding phonon-glass and electron-crystal thermoelectric properties. 3) In *Science* **298**, 2358-2361 (2002): *Kitaura et al.* reported the direct observation of dioxygen molecules physisorbed in the nanochannels of a microporous copper coordination polymer by the MEM (maximum entropy method)/Rietveld method, using in situ high-resolution synchrotron x-ray powder diffraction

measurements. In addition to the above examples, the reviewer may be referred to several other examples published in Nature Communications (e.g. *Nat. Commun.* **8**, 15152 (2017); *Nat. Commun.* **5**, 3515 (2014), etc.).

- d) From the MEM point of view, the result in this work is highly novel since it is, to our knowledge, the first application of the MEM in the direct monitoring of the temperature-driven static-dynamic transition of the atomic distribution in a simple crystalline solid.
- e) The 3D- Δ PDF technique is very new and only a handful of studies are available in the literature. It represents an exciting new frontier in structural research. As an example see *Nature* **578**, 256–260 (2020).
- f) From the theoretical part point of view, our work presents formation energies of intrinsic defects, 1D In^{1+} -ion migration pathways and energy barriers, and molecular dynamics simulation showing diffusion/hopping behavior of In^{1+} in InTe. Moreover, we have also shown that many other TlSe-type compounds have comparable or lower A^{1+} -ion migration energy barriers along the c axis than that of InTe, suggesting that the 1D disordered diffusion channel of the A^{1+} -ions along the c direction is likely to appear in many other TlSe-type compounds. All these are completely novel comparing with prior studies.

Based on the above points, we believe that the scientific content in our work represents a very significant advance over the earlier studies in terms of both novelty and fundamental insight into the structural disorder and ion diffusion of thermoelectric InTe and related TlSe-type compounds.

Comment 2

'1). Several times throughout the manuscript, the authors use the term "diffusion" to characterize the In^{1+} cations. It is however not clear whether this phenomenon remains localized in the cages formed by the octahedral chains or whether a truly ionic conduction takes place. According to prior studies, this mechanism has not been observed. This point should be clarified in the manuscript. In addition, it would be worthwhile to recall that ionic conductors cannot be considered as serious candidates for integration in thermoelectric generators due to the rapid deterioration of their performance this mechanism implies.'

Reply

Thank you for the comment. From MEM density, 3D- Δ PDF, and theoretical calculations, we have clearly revealed in the paper that the In^{1+} diffu-

sion/hopping along the c axis indeed happens in InTe. But the In^{1+} -ion diffusion/hopping probability is generally small as the MEM electron density value of the diffusion channel even at 700 K is about 5.3% of the maximum peak value of the main In^{1+} site. This essentially means that the ion diffusion/hopping in InTe is typically smaller than those in conventional superionic conductors. This might pose a great challenge to probe the signature of superionic conductivity. An external driving force such as a higher temperature, a large electric field, or the increased In^{1+} vacancy may be required to enhance the diffusion/hopping probability. Indeed, there are several reports proving the superionic conductivity signature by applying a large electric field in several other TlSe-type materials (e.g. $\text{TlInTe}_2/\text{TlInSe}_2$ (*Semiconductors* **45**, 1387-1390 (2011)), TlGaTe_2 (*Jpn. J. Appl. Phys* **50**, 05FC09 (2011)), etc.). It remains to be seen if InTe shows similar behavior. This would be a nice future project.

To make it clear, we have added one sentence on page 8: “The ion diffusion/hopping in InTe is clearly weaker than those in conventional superionic conductors^{5,9,29}, which might pose a challenge to probe the signature of superionic conductivity in InTe. An external driving force such as a higher temperature, a large electric field²⁰, or the increased In^{1+} vacancy may be required to enhance the diffusion/hopping probability.”

As the In^{1+} diffusion/hopping is not very significant comparing with typical superionic conductors, it is expected that the properties will not show rapid deterioration. However, systematic scientific work on stability during operating conditions need to be carried out in the future to determine whether InTe can be considered as serious candidates for integration in thermoelectric generators. It is too early to make any conclusion at this point. Also, the relevant discussion is clearly beyond the focus of this paper. We note that superionic conductors such as Cu_2Se and Zn_4Sb_3 are widely considered for implementation in thermoelectric devices even though they are challenged by stability issues.

Comment 3

'2). A comparison of the temperature dependences of the lattice thermal conductivity (Figure 1d) and of the electrical resistivity (Figure in the SI) with those reported previously (Ref. 22 in the manuscript) would be interesting. In particular, the dielectric maximum in the lattice thermal conductivity is observed near 4 K in the present study while it seems it was not yet reached at 2 K in Ref. 22. This may point to differences in crystalline quality, perhaps due to different vacancy concentration. To clarify this point, the hole concentration of the present crystal should be measured and discussed. This would allow a connection with the chemical composi-

tion reported ($\text{In}_{0.98}\text{Te}$) to be possibly made. More precisely, is the hole concentration consistent with this composition assuming one hole per In+vacancy? Moreover, in Ref. 16, much lower In vacancy concentrations have been suggested in polycrystalline samples. A detailed discussion about this point should be provided in the manuscript. Finally, the authors should also mention that the values of the thermal conductivity at high temperatures are similar in both crystallographic directions ($hh0$ plane and c direction) according to the results reported in Ref. 30, showing that the thermal transport in single-crystalline InTe is nearly isotropic.'

Reply

Thank you for the comment. After carefully checking the data reported in Ref. 22, we would like to point out that the thermal conductivity data in Ref. 22 was only measured down to 5 K and not 2 K as the reviewer mention in the comment (see the right panel of Figure A below). The authors in Ref. 22 also expect a maximum below 5 K as they mentioned that "the slight change in the slope observed near 5 K suggests that a maximum would probably be attained slightly below this temperature". In addition, in the left panel of Figure A we compare our thermal conductivity data with that reported in Ref. 22. The low-temperature thermal conductivity (as well as resistivity values of ~ 4.0 - 7.3 m Ω cm at 300-723 K) shows a fairly good agreement between our work and Ref. 22, indicating similar crystalline quality. Hence, it is not necessary to discuss the Hall carrier concentration. But we agree with the idea of comparing thermal conductivity data, and in the revised manuscript we have included in Fig. 1d the thermal conductivity data from Ref. 22 and 30 for comparison.

[Redacted]

Figure A. Thermal conductivities of InTe single crystal of this work in comparison with those reported in Refs 22 and 30 (*Phys. Rev. Res.* **2**, 043371 (2020); *J. Mater. Chem. C* **9**, 5250-5260 (2021)).

We do not think that it is meaningful to make a simple comparison between the hole concentration and the simple estimation from the composition $\text{In}_{0.98}\text{Te}$ by assuming one hole per In^+ vacancy. Strictly speaking, this kind of simple estimation is problematic. The dominant intrinsic defects include not only In^{1+} vacancy but also indium interstitials. Moreover, the charge transfer for In^{1+} in reality may not be complete. In principle, a comprehensive study on phase diagram calculation combined with defect chemistry and electronic density of states is required for estimating the carrier concentration from composition (e.g. see *Joule*, **2**, 141-154 (2018)). However, we are unable to conduct such calculations as the PBE functional applied in this work is not able to open a bandgap for InTe. More advanced functionals are required for such calculations, and this is beyond the scope of this paper.

The difference in carrier concentration data between polycrystalline and single-crystalline samples is not unexpected. Hall carrier concentration in chain-like single-crystalline InTe sample is generally anisotropic considering different band curvatures along different directions (e.g. see anisotropic carrier concentrations in single-crystalline SnSe, *Nature* **508**, 373-377 (2014)). In Ref. 22 Hall carrier concentration of the single-crystalline sample is measured along the c axis, whereas for the polycrystalline sample in Ref. 16 Hall carrier concentration is averaged over different directions considering random orientations of many grains. Moreover, the microstructures of single-crystalline and polycrystalline samples are usually different. Therefore, it is not unusual to see the difference in carrier concentrations between single-crystalline and polycrystalline InTe samples.

Regarding the last comment on nearly isotropic thermal conductivity, we have added one sentence mentioning the nearly isotropic feature of thermal conductivity at high temperatures in the revised manuscript.

To address this comment, the following revisions have been applied:

- a) Fig. 1d in the main text is updated by including reported experimental data from refs. 22 and 23. The ref. 30 is moved ahead as ref. 23 in the revised manuscript.
- b) Two sentences are added on page 4: “The measured low-temperature thermal conductivity data along the c direction show a very good agreement with those reported in ref. 22. By comparing with the reported data along the $[110]$ direction in ref. 23, it is found that the thermal conductivity of InTe single crystal is clearly anisotropic at room temperature but becomes less anisotropic or even nearly isotropic at high temperatures.”

Comment 4

'3). The kink observed in the electrical resistivity curve has been already observed and discussed in prior studies (Ref. 22 and S. Y. Back, *ACS Applied Energy Mater.* 3, 3628 (2020), this last study is not mentioned in the manuscript). The authors should mention that this kink is consistently observed in both single-crystalline and polycrystalline InTe at low temperatures and is thus not due to a measurement artifact. In addition, the large thermal displacement parameters of the In⁺ cations had been originally observed in the study of T. Chattopadhyay et al. (*J. Phys. Chem. Solids*, 46, 351 (1985)). This should be clearly mentioned in the present manuscript as well.'

Reply

Thank you for the comment. According to the suggestion by the reviewer, in the revised manuscript we have made the corresponding revision by mentioning that the resistivity kink is consistently observed in both single-crystalline and polycrystalline InTe at low temperatures and is thus not due to a measurement artifact. In addition, in the revised manuscript, we have included S. Y. Back, *ACS Applied Energy Mater.* 3, 3628 (2020).

However, we would like to point out that the reviewer's comment on "the large thermal displacement parameters of the In⁺ cations had been originally observed in the study of T. Chattopadhyay et al. (*J. Phys. Chem. Solids*, 46, 351 (1985))" is not quite true. In our original manuscript we cited the paper by Hogg & Sutherland (*Acta Crystallogr. B* 32, 2689-2690 (1976)) as the first report on refined ADPs, showing the large ADPs of In¹⁺ cations. This report was published much earlier than the reference mentioned by the reviewer. Therefore, we did not include the study of T. Chattopadhyay et al.. We have now included one sentence mentioning that the large ADPs of In¹⁺ were originally observed by Hogg and Sutherland (*Acta Crystallogr. B* 32, 2689-2690 (1976)).

To address the comment, we have done the following revisions:

- (a) One sentence is added on page 7: "The kink in resistivity data has been consistently observed in both single-crystalline²² and polycrystalline³¹ InTe at low temperatures and is thus not due to a measurement artifact."
- (b) One reference is added in the main text: "31. Back, S. Y. et al. Temperature-induced Lifshitz transition and charge density wave in InTe_{1-δ} thermoelectric materials. *ACS Appl. Energy Mater.* 3, 3628-3636 (2020)."

(c) One sentence is added on page 4: “Thermal displacement parameters of In^{1+} ions were found by Hogg and Sutherland²¹ to be very large and anisotropic with the maximum vibration along the c axis.”.

Comment 5

‘4). The authors mention in their manuscript that “lone-pair-induced strong anharmonicity, the interstitial, disordered, diffusive In^{1+} ions thereby should be considered as a key origin of the ultralow thermal conductivity as well as its weak temperature dependence in InTe .” This sentence is a bit misleading for readers since the link between the disordered In^{1+} cations and the lattice thermal conductivity has already been considered as indeed being a key origin in Ref. 22. This study has notably showed how this structural characteristic gives rise to low-energy optical modes with a low dynamic structure factor, as observed by inelastic neutron scattering. A more detailed discussion based on these results should be provided in the present manuscript.’

Reply

Thanks for the comment. In the revised manuscript, we have done corresponding revisions with a more detailed discussion based on ref. 22. Since the focus of the current work is on a systematic study of structural disorder in InTe and the current paper is already quite long, we do not show too many detailed discussions or investigations given the constraints imposed by journal style. We anticipate that the current systematic structural disorder study can provide a basis for a systematic theoretical study on the correlation between this structural disorder and thermal conductivity in a future project.

One new paragraph is added in the Discussion section on page 10: “The interstitial, disordered, diffusive In^{1+} ions should be considered as an important origin of the ultralow, weak temperature dependent lattice thermal conductivity in InTe . Correlated disorder is known to lead to the broadening of phonon linewidth that is inversely proportional to lattice thermal conductivity³⁸. Correlated disorder of indium vacancies and interstitials in 1D In^{1+} chains could potentially broaden the phonon linewidth for the low-energy In^{1+} vibration modes, resulting in the suppression of phonon lifetime, phonon mean-free path, and thereby lattice thermal conductivity. In particular, the structural disorder of In^{1+} ions was suggested by Misra et al.²² to be a key origin of the broadening of In^{1+} -weighted low-energy optical modes and the limited energy range of heat-carrying acoustic phonons, lowering the lattice thermal conductivity. Moreover, the plateau with nearly temperature-independent behavior

($\sim T^{-0.1}$) in thermal conductivity at $\sim 25-80$ K may be attributed to the correlated static disorder in 1D In^{1+} chains revealed by the MEM density and 3D- Δ PDF, similar to those observed in strongly disordered materials³⁹⁻⁴¹. With increasing temperature above the Debye temperature of ~ 120 K, as the intrinsic phonon-phonon scattering begins to be dominant thermal conductivity shows a clear decreasing trend but its temperature dependence is still a bit weaker than T^{-1} , which is likely induced by the wavelike tunneling^{11,14} contribution from interstitial, disordered, diffusive In^{1+} ions. However, the exact impact of the structural disorder with interstitial, disordered, diffusive In^{1+} ions comparing with the anharmonic phonon-phonon scattering on lattice thermal conductivity in InTe still requires further systematic theoretical and experimental investigations.”

All corrections are marked in red with the yellow background in revised manuscript.

Once again we would like to thank the reviewers for the time they have devoted to our manuscript. Their very insightful comments have significantly improved our manuscript and we look much forward to your decision.

REVIEWERS' COMMENTS

Reviewer #1 (Remarks to the Author):

The revised version of this manuscript addresses all my as well as other reviewers' concerns satisfactorily. I understand the length limitations in this journal has constrained the discussion involving disordered channel and thermal conductivity. However, I feel it would be better for the readers to appreciate the work if it is discussed in the supplementary notes. The manuscript, however, details all necessary observations that the reviewers asked and thereby I recommend it to be published in its current form.

Reviewer #2 (Remarks to the Author):

As a reviewer, I appreciate the efforts of the authors. The newly presented version has improved with respect to the previous one, and almost all the comments have been clearly addressed. Still, the following point should be attentional.

1. Although I understand the length and content of the paper are constrained by the journal style, the mechanism analysis on thermal properties is insufficient. I see authors add some discussions to explain the mechanism of how disordered In¹⁺ ions help in reducing thermal conductivity. However, some quantitative analysis of phonon properties specifically the phonon mean-free path mentioned in the paper is still required. I highly recommend authors do the phonon analysis such as calculating phonon-mean free path thereby to show direct evidences that disordered In¹⁺ ions causing low thermal conductivity of InTe, maybe in a future work. Molecular dynamic simulation may be a good tool to do this.

2. In addition, I suggest authors to add a short sentence to describe the background of "thermoelectric" in the manuscript, since the word is included in paper title.

Reviewer #3 (Remarks to the Author):

In this revised manuscript, the authors have well taken into account all the remarks of the three Reviewers. The changes made have improved the clarity and conciseness of several parts of the manuscript. Regarding its suitability for publication in Nature Communications, I understand the viewpoint of the authors. Nevertheless, I still consider that the present results may be a better fit to a more specialized journal. Although these results undeniably provide a nice experimental confirmation of the disordered character of the In⁺ cations, this hypothesis, previously envisaged in several prior studies on InTe, was very likely due to the rather simple crystal structure of InTe. This makes the results presented by the authors not really surprising or thought-provoking, as one may expect to find in Nature Communications papers. That said and given the positive reports made by the two other Reviewers, I do not have any further objection to the publication of this manuscript if the Editor considers it suitable for the wide readership of this journal.

Reviewer 1

Comment 1

'The revised version of this manuscript addresses all my as well as other reviewers' concerns satisfactorily. I understand the length limitations in this journal has constrained the discussion involving disordered channel and thermal conductivity. However, I feel it would be better for the readers to appreciate the work if it is discussed in the supplementary notes. The manuscript, however, details all necessary observations that the reviewers asked and thereby I recommend it to be published in its current form.'

Reply

We thank the referee for his/her careful evaluation of our response letter and recommendation for publication.

Reviewer 2

Comment 1

'As a reviewer, I appreciate the efforts of the authors. The newly presented version has improved with respect to the previous one, and almost all the comments have been clearly addressed. Still, the following point should be attentional.'

1. Although I understand the length and content of the paper are constrained by the journal style, the mechanism analysis on thermal properties is insufficient. I see authors add some discussions to explain the mechanism of how disordered In¹⁺ ions help in reducing thermal conductivity. However, some quantitative analysis of phonon properties specifically the phonon mean-free path mentioned in the paper is still required. I highly recommend authors do the phonon analysis such as calculating phonon-mean free path thereby to show direct evidences that disordered In¹⁺ ions causing low thermal conductivity of InTe, maybe in a future work. Molecular dynamic simulation may be a good tool to do this.

2. In addition, I suggest authors to add a short sentence to describe the background of “thermoelectric” in the manuscript, since the word is included in paper title.’

Reply

We thank the referee for his/her careful reading and comments. We appreciate the referee’s suggestion. Detailed investigation of the quantitative impact of structural disorder using a comprehensive theoretical model will be conducted and presented in a future work.

Regarding the comment on “thermoelectric”, we have now added a short sentence for introducing the background of “thermoelectric” in the first paragraph of the Introduction: “For the TE technology directly interconverting heat and electricity, reducing thermal conductivity is an essential strategy to improve the figure of merit $zT = \alpha^2 \sigma T / \kappa$ of a TE material, where α , σ , T , and κ represent the Seebeck coefficient, electrical conductivity, absolute temperature, and thermal conductivity, respectively.”

Reviewer 3

Comment 1

‘In this revised manuscript, the authors have well taken into account all the remarks of the three Reviewers. The changes made have improved the clarity and conciseness of several parts of the manuscript. Regarding its suitability for publication in Nature Communications, I understand the viewpoint of the authors. Nevertheless, I still consider that the present results may be a better fit to a more specialized journal. Although these results undeniably provide a nice experimental confirmation of the disordered character of the In⁺ cations, this hypothesis, previously envisaged in several prior studies on InTe, was very likely due to the rather simple crystal structure of InTe. This makes the results presented by the authors not really surprising or thought-provoking, as one may expect to find in Na-

ture Communications papers. That said and given the positive reports made by the two other Reviewers, I do not have any further objection to the publication of this manuscript if the Editor considers it suitable for the wide readership of this journal.'

Reply

We thank the referee for the comment and the time he/she has spent on our manuscript. In our previous response letter, we provided a very detailed explanation on the novelty and impact of the scientific content of our work. We gratefully invite constructive criticism, but it has to have a scientific basis. The novelty of a paper should be evaluated from the scientific content with rational scientific arguments rather than from personal opinion.

All corrections are marked with track changes in revised manuscript.

Once again we would like to thank the reviewers for the time they have devoted to our manuscript. Their very insightful comments have significantly improved our manuscript and we look much forward to your decision.